

# Aerosol-cloud interactions in mixed-phase convective clouds. Part 2: Meteorological ensemble.

Annette K. Miltenberger[1], Paul R. Field[1,2], Adrian A. Hill[2], Ben J. Shipway[2], and Jonathan M. Wilkinson[2]

[1]Institute of Climate and Atmospheric Science, School of Earth and Environment, University of Leeds, United Kingdom
[2]Met Office, Exeter, United Kingdom

*Correspondence to:* Annette K. Miltenberger (a.miltenberger@leeds.ac.uk)

**Abstract.** The relative contribution of variations in meteorological and aerosol initial and boundary conditions to the variability in modelled cloud properties are investigated with a high-resolution ensemble (30 members). In the investigated case, moderately deep convection develops along sea-breeze convergence zones over the southwestern peninsula of the UK. A detailed analysis of the mechanism of aerosol cloud interactions in this case has been presented in the first part of this study

(Miltenberger et al., 2017).

The meteorological ensemble (10 members) varies by about a factor 2 in boundary layer moisture convergence, surface precipitation, and cloud fraction, while aerosol number concentrations are varied by a factor 100 between the three considered aerosol scenarios. If ensemble members are paired according to the meteorological initial and boundary conditions, aerosol-induced changes are consistent across the ensemble. Aerosol-induced changes in CDNC, cloud fraction, cell number and size, outgoing

shortwave radiation, instantaneous and mean precipitation rates, and precipitation efficiency are statistically significant at the 5 % level, but changes in cloud top height or condensate gain are not. In contrast, if ensemble members are not paired according to meteorological conditions, aerosol-induced changes are statistically significant only for CDNC, cell number and size, outgoing shortwave radiation, and precipitation efficiency. The significance of aerosol-induced changes depends on the aerosol scenarios compared, i.e. for an increase or decrease relative to the standard scenario.

A simple statistical analysis of the results suggests that a large number of realisations (typically > 100) of meteorological conditions within the uncertainty of a single day needs to be consider for retrieving robust aerosol signals in most cloud properties. Only for CDNC and shortwave radiation small samples are sufficient.

While the results are strictly only valid for the investigated case, the presented evidence combined with previous studies highlights the necessity for careful consideration of intrinsic predictability, detailed meteorological conditions, and co-variability

between aerosol and meteorological conditions for observational or modelling studies of aerosol indirect effects.





# 1 Introduction

Clouds and precipitation are an integral part of the atmospheric system relevant for weather and climate. Considerable uncertainty remains in our understanding and modelling of clouds and their interaction with other parts of the climate systems. Main issues are an incomplete physical understanding of cloud microphysical processes, a lack of quantitative formulations

representing microphysical processes on the several orders of magnitude larger model grid scale, and the many non-linear interactions between different components of the system. In the last decades, in particular the modification of cloud properties by aerosols has been studied due to the large increase in anthropogenic aerosol emissions and the relation between particle number concentrations and radiation.

Many modelling studies have investigated the impacts of an aerosol change on either isolated clouds or larger cloud fields,

but found different responses of the studied clouds depending on environmental conditions, model formulations, duration of simulations, and domain size (recent reviews by Tao et al., 2012; Altaratz et al., 2014; Rosenfeld et al., 2014; Fan et al., 2016). Recent studies have highlighted that it is necessary to simulate entire cloud fields over long periods in order to quantify a climate relevant aerosol signal (e.g. Grabowski, 2006; van den Heever et al., 2011; Seifert et al., 2012). This is necessitated by interactions between clouds and their thermodynamic environment. These interactions can at least partly compensate the large

changes to simulated for individual clouds. For example, Seifert et al. (2012) showed that while large local changes in precipitation occurred, mean precipitation did not change significantly in perturbed aerosol simulations of summer time convection over Germany.

The highly non-linear nature of convective cloud dynamics and microphysics also calls for the use of large ensembles due to a potentially rapid growth of small perturbations to the system (e.g. Wang et al., 2012). Therefore, it is vital to understand the in-

herent predictability limits of the investigated cloud system to retrieve robust aerosol-cloud interaction signals (e.g. Grabowski et al., 1999; Khairoutdinov and Randall, 2003; Morrison and Grabowski, 2011). While the importance of predictability aspects has been acknowledged in weather forecasting, its implications for the evaluation of cloud microphysics parameterisations or the quantification of aerosol-cloud interactions has only been acknowledge in a few studies (Grabowski et al., 1999; Khairoutdinov and Randall, 2003; Zeng et al., 2008; Morrison and Grabowski, 2011; Morrison, 2012). To our knowledge, the first

study to highlight the importance of intrinsic predictability for cloud microphysics evaluation and aerosol-cloud interactions is Grabowski et al. (1999). Along with changes to various parameters in the cloud microphysics, cloud-radiation interaction, and CCN number concentrations, they applied random perturbations to the large-scale forcing, the surface fluxes, and nudging timescale in their 2D simulations of deep tropical convection. Khairoutdinov and Randall (2003) investigated the sensitivity of convective clouds over the ARM southern great plains site to the choice of cloud microphysical parameterisations and per-

turbed initial conditions. While they found the mean hydrometeor profile and cloud fraction to be strongly dependent on the chosen cloud microphysical scheme, the variability of cloud fraction, precipitable water, and surface precipitation induced by different microphysical schemes was similar to those resulting from perturbed initial conditions. In a similar modelling framework to Grabowski et al. (1999), Morrison and Grabowski (2011) also applied random perturbations to simulations of deep tropical convection based on the Tropical Warm Pool International Cloud Experiment. They found a large variability in top of





atmosphere radiative fluxes between ensemble members generated by modest perturbations to the boundary layer temperature structure. Therefore a large ensemble with 240 members was required to retrieve a robust aerosol-induced signal in the top of atmosphere radiative fluxes. In their ensemble, surface precipitation was insensitive to aerosol changes. The simulations in these studies use large-scale forcing time series, which provide realistic time variations in forcing but do not allow for a

two-way interaction of the clouds with the large-scale forcing. While this avoids the even larger complexity of cloud-induced changes to large-scale forcing, it is ultimately necessary to include this interaction in order to quantify the impact of uncertainties in cloud microphysical processes or changes in aerosol concentration on the atmospheric system.

The relative importance of meteorological and aerosol conditions for cloud properties has also implications for obtaining observational evidence of aerosol-cloud interactions. Many observational studies of aerosol-induced changes in cloud properties

need to relay on correlations between bulk parameters (e.g. Devasthale et al., 2005; Koren et al., 2010; Gryspeerdt et al., 2014), which rises the question of co-variability and coincidence (e.g. Stevens and Feingold, 2009; Gryspeerdt et al., 2014). The importance of cloud dynamics in observational data-set has been recently demonstrated by Sena et al. (2016). The study analysed the correlation of aerosol, cloud dynamics, and a range of cloud properties for shallow warm-phase clouds over the ARM southern Great Plains site. They showed that the variability of cloud radiative properties was dominated by cloud dynamics

rather than cloud microphysical properties.

One approach to investigate the role of intrinsic predictability and the relative importance of aerosol and meteorological variability is the use of convection-permitting ensemble systems. Ensemble forecasting is now an important component of operational forecasting and is increasingly used at convection-permitting or even higher spatial resolutions (e.g. Bowler et al., 2008; Marsigli et al., 2014; Beck et al., 2016). The use of convection-permitting ensemble forecasts provides a means for

assessing the magnitude of aerosol-induced changes in the context of variations in the cloud and precipitation evolution to perturbations in the meteorological conditions, which are consistent with the available meteorological observation uncertainty. Besides offering insight into the questions of robustness and observability of aerosol- induced changes, the ensemble approach explores whether perturbations of the aerosol environment should be included in the future forecasting systems for quantitative precipitation forecasts.

In the present study, we investigate the robustness and relative importance of aerosol-induced changes in mixed-phase, sea-breeze related convective cloud in high-resolution ensemble simulations with perturbed meteorological and aerosol initial and lateral boundary conditions. The case was selected from the COnvective Precipitation Experiment (COPE) that was conducted over the southwestern peninsula of the UK in 2013 (Blyth et al., 2015; Leon et al., 2016). On the selected day ($3^{\text{th}}$ August 2013) deep convective clouds with maximum cloud top heights of about $5\,\mathrm{km}$ developed in the late morning along converging

sea-breeze fronts. The line of convective clouds remained roughly stationary along the main axis of the peninsula until the late afternoon. Generally, new cells formed at the southwestern tip of the peninsula and merged into larger cloud clusters while propagating northeastwards along the line. Simulations of this case were conducted with the Unified Model (UM) at a spatial resolution of $250\,\mathrm{m}$ using the newly developed Cloud-AeroSol Interacting Microphysics Module CASIM (Shipway and Hill, 2012; Hill et al., 2015; Grosvenor et al., 2017; Miltenberger et al., 2017). The comparison of the base-line simulation

with observational data and the sensitivity of cloud properties to aerosol perturbations was presented in the first part of this





study (Miltenberger et al., 2017) and is briefly summarised here: Increasing aerosol concentrations suppress precipitation in the morning. With progressing organisation of the clouds along the sea-breeze fronts, the response transitions into a precipitation enhancement. In the early phase, precipitation decreases continuously with aerosol concentration (0.1 to 30 times observed value), while in the afternoon the largest accumulated precipitation occurs with observed aerosol profile. The limitations on

cloud deepening from a mid-tropospheric stable layer were hypothesised to inhibit a further increase of precipitation for aerosol number concentrations larger than the observed values. Vertical velocities increase in the convective core regions with aerosol concentrations. However, contrary to the convective invigoration hypothesis changes in latent heat release are dominated by changes in the warm-phase part of the cloud with very small changes above the $0\,°C$ line. It was hypothesised that accompanying changes in the cloud field structure (fewer, larger cells with increasing aerosol) were important for the changes in latent

heat release from condensation.

In this paper, we extend the analysis of Miltenberger et al. (2017) by including simulations with perturbed meteorological conditions in the analysis. With the combined perturbed meteorology and aerosol initial condition ensemble we investigate if the aerosol-induced changes are (i) robust to and (ii) significant relative to small changes in the meteorological initial conditions. The paper is structured as follows: Section 2 provides details on the model set-up and observational data used in this study. The

ensemble simulations are compared to observational data in section 3. In section 4, we discuss the variability of cloud properties in the perturbed meteorology only ensemble, while the impact of aerosol perturbations on clouds and precipitation for individual ensembles members is assessed in section **??**. Finally, the results from the full ensemble, i.e. including perturbations to meteorology and aerosols, are presented in section 6. The findings are summarised in section 7.

## 2   Model and data

The initial condition ensemble discussed in this paper is constructed by downscaling 9 ensemble members from the operational global ensemble system of the Met Office (MOGREPS, Bowler et al. (2008)) over the southwest peninsula of the UK. The global model ensemble is recomputed from the Met Office operational analysis for 18 UTC on 02. 08. 2013 and perturbations to the initial conditions with the global model version and set-up used for the operational forecast in 2013 (UM, vn8.2, PS31 configuration, N400 resolution, i.e. $\approx\,33\,km$ in mid-latitudes). Stochastic physics as described in Bowler et al. (2009) are used.

9 members are selected for the global ensemble for dynamical downscaling with higher resolution regional simulations. For the selection, the time-series of moisture and moist static energy convergence over the regional model domain are computed (Fig. 1) and 9 members are selected with a hierarchical clustering algorithm. A similarity matrix is constructed by summing the Euclidean distances of moisture convergence and static energy from hourly model output over the 24 hour time series. 9 clusters are then defined with the algorithm by J. H. Ward (1963). For each cluster the member closest to the mean cluster time

series is chosen for downscaling. Note, that while this procedure provides a sampling of different time series, it does not necessarily retain the statistical properties of the global ensemble. It is also known that convective-permitting ensembles constructed by downscaling global ensemble members do not represent the meso-scale error characteristic correctly (e.g. Saito et al., 2011; Berner et al., 2011). Convection-permitting ensemble forecasts are often underdispersive (e.g. Romine et al., 2014; Schwartz



et al., 2014). The downscaled ensemble will represent some unknown fraction of the true meteorological uncertainty for the studied day. Due to the cluster selection and the downscaling approach, the meteorological uncertainty is likely underestimated in the current study. Although the ensemble selection and initialisation of the ensemble should be improved in future studies, we do not think that this is a strong caveat to the main conclusions of the current study.

The selected members of the global ensemble are dynamically downscaled, i.e. provide the initial and lateral boundary conditions for two nested regional simulations. In addition, to the 9 ensemble members, we use simulations driven by the unperturbed global forecast. These are referred to as "control", but are included in the term "ensemble members" if not stated differently. The ensemble members have been sorted according to the large-scale moisture convergence computed from the fluxes at the domain boundary: ensemble member 1 has the largest large-scale moisture convergence and ensemble member 9 the smallest.

Regional simulations with a grid spacing of $1\,\mathrm{km}$ (500 by 500 grid points) are started at $00\,\mathrm{UTC}$ on 03. 08. 2013. These simulations provide the initial and boundary conditions for simulations in a second set of nested simulations with a h a grid spacing of $250\,\mathrm{m}$ (900 by 600 grid points). For the regional simulations, we use the UM version 10.3 (GA6 configuration, Walters et al. (2017)) with the Cloud-AeroSol Interacting Microphysics module (CASIM) (Shipway and Hill, 2012; Hill et al., 2015; Grosvenor et al., 2017; Miltenberger et al., 2017). The model set-up for the regional simulations is identical to the set-up

described in Miltenberger et al. (2017). The control simulations are identical to the simulations used in Miltenberger et al. (2017) with the only difference that the simulations discussed here use the cloud droplet number predicted by CASIM instead of a prescribed value for the computation of the radiative fluxes. In all regional simulations, moisture conservation is enforced according to Aranami et al. (2014, 2015).

Cloud microphysical processes are parameterised with the CASIM module which is a double-moment microphysics scheme

with five different hydrometeor categories. The CASIM module can represent the interactions between aerosol fields and cloud microphysical properties. For the ensemble simulations, we use the so-called "passive aerosols" mode: aerosol fields are used for droplet activation and ice nucleation, but are not altered by cloud microphysical processes. The impact of this choice on the representation of aerosol-cloud interactions is discussed in Miltenberger et al. (2017). Aerosol initial and lateral boundary conditions are derived from aircraft data as described in Miltenberger et al. (2017). In the following, simulations with the aerosol

profile derived from observations are referred to as "standard aerosol" simulations. Additional simulations of each ensemble member are performed with perturbed aerosol profiles, for which aerosol number densities and mass mixing ratio are multiplied at all altitudes by a factor 10 ("high aerosol") and 0.1 ("low aerosol"), respectively. The mean aerosol radius is conserved in the perturbed profiles. Accordingly, the entire ensemble with perturbed meteorological and aerosol initial conditions has 30 members in total.

For the evaluation of the ensemble with the standard aerosol profile, we use the same set of observations as in the first part of this study. These include radiosonde and aircraft data from the COPE field campaign and data from the operational radar network. Details about these data-sets can be found in Miltenberger et al. (2017).



## 3 Evaluation of ensemble simulations

### 3.1 Radar reflectivity and surface precipitation

In all ensemble simulations, a convergence line develops roughly over the centre of the Peninsula in the early afternoon (SI Fig. 1). The overall meteorological evolution is similar to the control simulation. The domain-average precipitation increases

in all ensemble members during the morning hours and reaches maximum values between 13 UTC and 16 UTC (Fig. 2a). Domain-average precipitation rates from the control forecast (dashed blue line) are mostly within the spread of the ensemble members (blue shading), although the ensemble mean domain-average precipitation rate is about a factor 2 smaller than the control during the period of main convective activity (12 -17 UTC). The spread of the ensemble including aerosol perturbations (cyan shading) is not much larger than the ensemble spread based on perturbed meteorological conditions alone, particularly

after about 1430 UTC. The ensemble mean is almost identical for both ensembles. The domain-average precipitation rates derived from radar (Radarnet IV, Harrison et al. (2009); MetOffice (2003)) are mostly outside the spread of the ensemble. This indicates that either the ensemble is underdispersive or that there are issues with the radar derived surface precipitation eventually related to assumptions on hydrometeor properties and sub-cloud evaporation in the retrieval algorithm (e.g. Li and Srivastava, 2001). Previous evaluation studies of convection-permitting ensemble simulations have reported precipitation fore-

casts to be underdispersive also over longer evaluation periods (e.g. Romine et al., 2014; Schwartz et al., 2014). This indicates that the ensemble does not include all sources of uncertainty, e.g. due to structural or parametric uncertainty in the model physics, and/or that perturbations to the initial and boundary conditions are not fully representative of the true uncertainty. The incorporation of perturbations to the aerosol initial and boundary conditions does not improve the comparison strongly. The underdispersivity of the ensemble does not strongly impact the major conclusions of our study, as we interpret the meteorolog-

ical uncertainty as a lower limit of meteorological variability in the discussion (section 7).

The underestimation of domain-average precipitation in the ensemble is due to a combined underestimation of precipitating area fraction and the occurrence frequency of medium precipitation rates in precipitating areas. The observed domain-average in-cloud precipitation is within the ensemble spread, but in the lower range of the predicted values (SI Fig. 2 a). Combined with the underestimation of the domain-average precipitation rate (Fig. 2a), this indicates too few grid points with surface pre-

cipitation. In addition, the distribution of precipitation rates in the rainy part of the domain shows a too infrequent occurrence (spatial or temporal) of medium rain rates ($0.2 - 4\,\mathrm{mm\,h^{-1}}$) in the model data (SI Fig. 2 b). Overall this is very similar to the performance of the control simulation. While including the aerosol perturbations in the initial conditions increases the spread of precipitation rates, the ensemble spread does not cover the radar derived precipitation rate distribution for the ensemble with either meteorology or meteorology and aerosol perturbations.

Differences between the observational data, the control simulation, and the average of the ensemble simulations are much smaller, if in-cloud low-level radar reflectivity is compared directly (Fig. 2b). The ensemble mean distribution of radar re-flectivity at 750 m and the control simulation agree within $\pm 1\,\mathrm{dBZ}$ for a reflectivity larger than $15\,\mathrm{dBZ}$ and within $\pm 5\,\mathrm{dBZ}$ for a smaller reflectivity values. Compared to the observational data deviations in the mean are smaller than $\pm 5\,\mathrm{dBZ}$, except for the reflectivity range between $20 - 30\,\mathrm{dBZ}$. In this range, the observational occurrence frequency is higher than in any



member. As discussed in Miltenberger et al. (2017), the better agreement of the simulated and observed reflectivity distribution at 750 m than that of the precipitation rate distributions may indicate issues with the rain rate retrieval from the radar data due to differences in assumed and modelled sub-cloud evolution of the hydrometeor size distribution. In addition, there may be issues with the model derived reflectivity values due to the simple Rayleigh scattering assumptions.

The distribution of column maximum reflectivity in cloudy areas of the domain matches the observed distribution within 5 dBZ for values larger than 10 dBZ (SI Fig. 2 c). Aerosol perturbations significantly contribute to the spread in the distribution: The observational distribution is contained within the spread of the perturbed aerosol ensemble, but not in the spread of the ensemble using only perturbed meteorological initial and boundary conditions.

  The 3D radar composite available for this case provides information about the vertical structure of the clouds. Here we com-
pare the simulated and observed altitude of the highest occurrence of a radar reflectivity larger than 18 dBZ. This threshold is frequently used in radar products (e.g. Lakshmanan et al., 2013; Scovell and al Sakka, 2016). While the control simulation has slightly higher values in the morning hours ($200 - 500$ m for $11 - 13$ UTC), the ensemble mean is closer to the observed evolution (within 200 m, SI Fig. 3 a). The inclusion of perturbed aerosol initial and boundary conditions has only a small impact on the ensemble mean height of the 18 dBZ contour (maximum difference: $\pm 100$ m), except for the last one (two) hours,
during which the observed height is within the spread of the ensemble without (with) perturbed aerosol initial conditions. Also for other reflectivity thresholds ($5 - 25$ dBZ), the observed mean height is within the ensemble spread and the difference to the ensemble mean is generally smaller than 500 m (not shown).

## 3.2   Radiosonde data

Thermodynamic profiles are available at two hourly intervals from radiosondes released at Davidstow ($50.64\,°$ N, $4.61\,°$ W).
These profiles are compared to the thermodynamic structure of the closest model grid column from the simulation with the standard aerosol profile (SI Fig. 4). The overall (out of cloud) structure of the temperature and dew-point temperature profiles are similar for all times and ensemble members. Temperature differences to the radiosonde profile are within $-1.7$ K $(-1.8$ K) to $+2.5$ K $(+2.9$ K) and dew-point temperature within $-8.8$ K $(-15.4$ K) to $+8.0$ K $(+11.6$ K) for all levels at 10 UTC (15 UTC), i.e. for observational data not compromised by the presence of a cloud. The spread is naturally larger for the
ensemble than for the control simulation. For the control simulation, temperature differences to the observed profile are smaller than 1 K and the dewpoint temperature agrees within 5 K (10 K) below (above) the stable layer. The observed cloud dew-point temperature generally falls within the spread of the ensemble members, with the exception of a lower observed humidity below 900 hPa at 15 UTC. The observed temperature also generally falls within the ensemble spread except for colder observed air temperatures in the layer between $550 - 400$ hPa. All ensemble members have a stable layer between $5 - 6$ km altitude,
which is an important determinant for the cloud top height (Miltenberger et al., 2017). The main differences between individual ensemble member is the humidity above 600 hPa, with the altitude of the driest point in this layer varying by about 100 hPa. The height of the $0\,°$ level and the lifting condensation level from the profiles corroborate the good agreement between observed and modelled profiles for the duration of the simulation and all aerosol scenarios: maximum deviations are about 300 m for the $0\,°$ level height and 400 m in the lifting condensation level (SI Fig. 3 b and c). While the ensemble mean lifting condensation



level is almost identical to the one in the control run, the ensemble mean height of the $0\,^{\circ}$C line is about $100\,\mathrm{m}$ higher than in the control run during the considered time period. As a result, the ensemble mean $0\,^{\circ}$C line is about $200\,\mathrm{m}$ higher than the altitude indicated by the radiosonde data. The ensemble spread does not incorporate the observed $0\,^{\circ}$ level height (except for the data point at 1350 UTC, during which the radiosonde passed through clouds. The lifting condensation level is within the

ensemble spread except for the observation at 1520 UTC.

Overall the ensemble reflects the cloud and precipitation evolution, and thermodynamic structure as indicated by observational data. However, the ensemble does not improve on the performance of the control run. Overall the ensemble performance provides confidence that the important physical mechanisms are well enough represented to conduct aerosol perturbation experiments.

## 4  Meteorological initial condition ensemble with unperturbed aerosol profile

All ensemble members form a line of convective clouds over the Peninsula, which is associated with convergence along the sea-breeze fronts. The column maximum reflectivity in the different members of the meteorological ensemble (using the standard aerosol profile) are shown in SI Fig. 1 for 14 UTC. However, the members vary in the amount of clouds and there are some differences in the location of the main cloud line and its orientation. These differences are not specific to the time instance

shown in SI Fig. 1, but persist throughout the simulations. Given the overall similar meteorological situation in the ensemble members, i.e. line of convective clouds forming along sea-breeze fronts, the main impact of the perturbed meteorological initial conditions should be (i) perturbations to vertical lifting and hence condensation, and (ii) the vertical cloud structure by modifications to the vertical wind shear and the thermodynamic limitations on cloud depth. The first can be quantified by the condensation ratio CR, which is the fraction of the incoming moisture flux that condenses in the domain. CR varies between

0.04 and 0.095 with the largest values occurring in ensemble members 1 and 2 followed by the control simulation and ensemble member 5. Section 4.1 will discuss differences in the upstream profiles in more detail, while section 4.2 describes the impact on the cloud field and precipitation formation. The discussion in this section focusses on the meteorological ensemble with the standard aerosol profile. The figure references in this section pertain only to the dark blue symbols representing simulations with the standard aerosol scenario.

### 4.1  Differences in upstream profiles and sea-breeze strength

Clouds develop mainly along the sea-breeze fronts propagating inland on the southwestern peninsula of the UK. The convergence induced by the sea-breeze circulation together with the large-scale moisture convergence and local buoyancy determines the domain-wide lifting and cloud formation. The main controlling factors for sea-breeze strength are the temperature difference between sea and land, the large-scale wind direction relative to the coastline, and the background wind speed (e.g.

Estoque, 1961; Miller et al., 2003). Stationary convergence lines over the southwest peninsula of the UK and associated convective activity have been shown to be sensitive to the differential heating of the land surface and the interaction with the





background wind field (Golding et al., 2005; Warren et al., 2014). The variation of these meteorological parameters across the ensemble members is shown in Fig. 3 for specific height levels, while the profiles of the wind components, temperature, and specific humidity are shown in SI Fig. 5.

The temperature difference between land and sea is $1 - 1.5\,\mathrm{K}$ in the morning and increases to $2\,\mathrm{K}$ by noon with the exception

of ensemble members 1 and 2, which have a smaller increase in the temperature difference (Fig. 3c, SI Fig. 6). These members have a higher cloud fraction (Fig. 6c), therefore radiative heating of the land surface is smaller and the land-sea temperature difference remains smaller. The variations in land-sea temperature gradients are most important in the morning, i.e. before 12 UTC, as the sea-breeze fronts develop at around 10 UTC. The differences developing later between the ensemble members should not strongly impact the sea-breeze convergence. The integrated convergence of the $10\,\mathrm{m}$ windspeed over the peninsula

(referred to as "low-level" convergence) is used as an indicator of the sea-breeze strength (Fig. 3 d, SI Fig 6 a). The low-level convergence does not correlate well with the land-sea temperature difference suggesting that the small variations between ensemble members ($\approx 0.5\,\mathrm{K}$) are not important for differences in convergence.

The wind speed in the boundary layer varies between about $8\,\mathrm{m\,s^{-1}}$ and $11\,\mathrm{m\,s^{-1}}$, and increases to values of $13 - 18\,\mathrm{m\,s^{-1}}$ at $4\,\mathrm{km}$ altitude. The wind direction is generally from the south-west with a variability of about $10\,^{\circ}$ and a shift towards a

more easterly direction at higher altitudes. Despite the variability between ensemble members in these variables, there is no clear relation of the wind speed or direction in the boundary layer to the integrated low-level convergence over the peninsula (Fig. 3). However, ensemble members with a smaller wind speed and a more southerly wind direction at $2 - 4\,\mathrm{km}$ altitude tend to have a stronger low-level convergence, i.e. the control and members 1 and 2.

Other important variables in the initial conditions for cloud and precipitation formation are the temperature and moisture pro-

files (SI Fig. 5 c and d). The temperature structure in all ensemble members is very similar with a well-mixed boundary layer below $800 \pm 200\,\mathrm{m}$, an almost moist-adiabatic temperature gradient up to $500\,\mathrm{hPa}$, and a layer of almost constant temperature between $500\,\mathrm{hPa}$ and $450\,\mathrm{hPa}$. This layer of constant temperature was found to be important for the vertical evolution of the clouds in the first part of this study. The point of the smallest temperature gradient in the troposphere ($-5.0$ to $-2.4\,\mathrm{K\,km^{-1}}$) is located between $4.1 - 5.5\,\mathrm{km}$ at $12\,\mathrm{UTC}$ in the grid column closest to Davidstow. As a result of the small variation in

the temperate profile, the average and maximum CAPE values are similar for all ensemble members ($100 - 160\,\mathrm{J\,kg^{-1}}$, SI Fig. 7 b). The moisture content in the boundary layer is $9 \pm 0.5\,\mathrm{g\,kg^{-1}}$ and differences between ensemble members remain smaller than $0.5\,\mathrm{g\,kg^{-1}}$ for all altitudes. Due to the decrease of moisture content with altitude, these variations of the moisture content become more important with altitude. The altitude of the driest point in the profile varies by about $100\,\mathrm{hPa}$ between ensemble members ($500 - 400\,\mathrm{hPa}$ at 10 UTC and $450 - 570\,\mathrm{hPa}$ at 14 UTC, SI Fig. 4). The driest point in the profiles is

generally very close to the altitude of the layer with constant temperature and therefore also contributes to the limitation of cloud top heights.

## 4.2 Differences in cloud field structure, condensate budget, and surface precipitation

The different meteorological initial and boundary conditions impact the low-level convergence in the domain, the thermodynamic profile, the wind shear, and the moisture influx and profile as discussed in section 4.1. These changes in the meteorolog-



ical conditions impact cloud formation, cloud field structure, and precipitation formation.

An important variable relating clouds and dynamics is the amount of lifting in the domain, which is influenced by convergence in the large- and meso-scale flow field as well as local buoyancy terms. The integrated condensate gain G, i.e. the integrated condensation and deposition rate, shows significant differences across ensemble members (Fig. 7a). G is very well correlated

with the net moisture flux at the top of the boundary layer (Fig. 4a, blue symbols), which is diagnosed from the sum of the moisture flux at the domain boundaries (red symbols) and the surface moisture flux (green symbols). This indicates that the domain-integrated lifting is primarily controlled by variations in lateral moisture convergence (based on the moisture fluxes at the domain boundaries), while variations in the domain internal, i.e. meso-scale, convergence are small. The surface moisture flux adds some modifications to the boundary layer moisture budget, e.g. ensemble members 3 and 4 and member 7 and 8,

respectively. A decomposition of the boundary layer top moisture export suggests that variations in the vertical velocity are more important than changes in the specific humidity (not shown). Ensemble members 4 and 7 have a particularly large surface sensible heat flux, likely increasing the boundary layer top moisture export over that expected from low-level convergence. The ratio of the incoming moisture flux that condenses in the domain (the condensation ratio CR) reflects the large-scale boundary layer convergences (Fig. 7b). It is not influenced by the different surface moisture fluxes as these are accounted for in the

moisture flux into the domain.

The cloud field structure is described in terms of the cloud fraction, cell number and mean size, and cloud top height. Cells are defined as coherent areas with a column maximum radar reflectivity larger than $25 \, \mathrm{dBZ}$. Cloud top height is defined by the highest model level with a condensed water content larger than $1 \, \mathrm{mg \, kg^{-1}}$. Cloud fraction is calculated as the arial fraction of the domain with condensed water path larger than $0.001 \, \mathrm{kg \, m^{-2}}$. The cloud fraction and cell number show variations

between ensemble members similar to those in condensate generation. Variations in mean cell size are quite small (Fig. 6 a - c). Mean cloud top height varies by about $750 \, \mathrm{m}$ between ensemble members (Fig. 6d) with largest (smallest) values for ensemble member 8 (4). The variation in mean cloud top height is in general consistent with the variations in the pressure of the equilibrium level between ensemble members (SI Fig. 7 c). The equilibrium level was determined from the grid-column closest to the Davidstow observational site. The distribution between low, medium and high cloud tops shows an about $20 \, \%$

difference between the ensemble members (SI Fig. 8 a). Ensemble members 4 and 7, which have high surface moisture fluxes, have the largest low-cloud fraction, while ensemble members 1, 2, 5, and 8 have a relatively large fraction of deep clouds.

Accumulated surface precipitation varies by about a factor 1.5 between ensemble members (Fig. 7c). Variations in precipitation efficiency PE (the ratio of surface precipitation to condensate gain) are less than $5 \, \%$ suggesting that the differences in precipitation are mainly due to changes in condensate generation, i.e. air mass lifting. Ensemble member 4 has a significantly

lower PE than the other ensemble members, which is likely related to the high fraction of shallow clouds with cloud tops below $2.5 \, \mathrm{km}$ and a therefore small contribution of mixed-phase processes to domain-wide precipitation formation. Conversely, ensemble member 8 has a relatively large PE and the largest fraction of clouds with tops above $4.3 \, \mathrm{km}$. The relatively small differences in precipitation efficiency are consistent with the almost invariant cloud droplet number concentrations for all ensemble members (Fig. 5). Mean surface precipitation rates as well as percentiles behave similar to variations in accumulated

precipitation between ensemble members (Fig. 8). Variations in the mean condensed water path are consistent with variations





in the condensate generation between ensemble members (Fig. 9).

Mean reflected shortwave radiation ranges from $130\,\mathrm{W\,m^{-2}}$ to $155\,\mathrm{W\,m^{-2}}$ (Fig. 11a). Largest (smallest) values occur for ensemble member 2 (7), which are the ensemble members with the largest (smallest) cloud fraction. Differences in outgoing longwave radiation are smaller than $3\,\mathrm{W\,m^{-2}}$ and are in general consistent with the distribution of cloud top heights (Fig. 11b).

## 5 Aerosol-induced changes in cloud properties for ensemble members with identical meteorological initial and boundary conditions

The simulation of each meteorological ensemble member was conducted with three different aerosol profiles: a so-called standard aerosol profile, which was derived from aircraft observations, and "low" and a "high" aerosol scenarios, which have a factor 10 lower respectively higher aerosol number concentration than standard aerosol profile. The modification of the aerosol number concentration is applied at all altitude. The mean and effective radius of the aerosols is identical in all aerosol profiles. The impact of the perturbed aerosol profiles on cloud and cloud field properties as well as precipitation formation in the control simulation has been discussed in the first of this study. Here we discuss aerosol-induced changes for individual meteorological ensemble member to establish whether the aerosol signal is consistent for slightly different meteorological initial and boundary conditions. Figure references pertain to the differences between the different coloured symbols for each ensemble member separately.

### 5.1 Cloud droplet number concentration

The cloud base and cloud top CDNC are shown in Fig. 5 for all ensemble members and aerosol profiles. All ensemble members show a consistent increase of the cloud base CDNC by about a factor 7 between the low (standard) and the standard (high) aerosol scenarios. The aerosol-induced change in cloud top CDNC is similar in all meteorological ensemble members with a change by about a factor 5.5 for each factor 10 increase in the background aerosol concentrations. The small differences between ensemble members suggest only minor changes in the cloud-base vertical velocity distribution.

### 5.2 Cloud field structure

The cloud field structure is described in terms of cloud fraction, cell number and size, and cloud top height (Fig. 6). For all ensemble members, the number of cells decreases with increasing background aerosol concentrations, but the cell area increases. The changes in these two variables largely compensate each other, so that the cloud fraction displays little sensitivity to the aerosol scenarios with changes being smaller than $0.01$. Although small, the aerosol-induced change in cloud fraction is consistent across all ensemble members.

The mean cloud top height is shown in Fig. 6d. The distribution of cloud top heights is represented by the fraction of cloud tops with altitudes below $2.5\,\mathrm{km}$, between $2.5\,\mathrm{km}$ and $4.3\,\mathrm{km}$, and larger than $4.3\,\mathrm{km}$ in SI Fig. 8 a. In all ensemble members, the mean cloud top height increase from the low to the standard aerosol scenario. The increase in mean cloud top is due to an increase in the fraction of cloud tops higher than $4.3\,\mathrm{km}$. The associated reduced fraction of clouds with lower cloud tops is



distributed differently over the low and medium cloud top height categories in the different ensemble members (change in low cloud top fraction dominant for ensemble members 1, 2, 7, 8, and9). For a further increase in aerosol number concentration, the mean cloud top height does not increase further (members 1 and 2) or even decreases (members 4, 5, 6, 7, 8, 9). The decrease in mean cloud top height for the latter is mainly due to an increase in the fraction of clouds with low cloud tops. The

fraction of clouds with cloud tops higher than $4.3\,\mathrm{km}$ is shows only very small changes between the simulations with standard and high aerosol profiles. The small change in cloud top height for increasing the aerosol number concentration beyond the standard aerosol scenario, is likely related to the presence of a mid-trosposheric stable layer, which is present in all ensemble members and limits cloud depths (section 4). Most larger convective cells have reached this "maximum" cloud top height for the standard aerosol scenario. The aerosol scenario has a very small impact on cloud base height with changes in mean cloud

base height less than $50\,\mathrm{m}$ (SI Fig. 8 b).

### 5.3   Condensed water budget and precipitation formation

The condensate gain G displays only little change for the different aerosol scenarios (Fig. 7a). G increases in most ensemble members by $0-4\,\%$ from the low to the standard aerosol scenario, while changes between the standard and high aerosol scenario amount to $-4-2.5\,\%$. The changes in condensate generation correspond to very small changes in the condensation

ratio (Fig. 7b). As discussed in Miltenberger et al. (2017), the asymmetry in the response to increased and decreased aerosol concentrations is likely related to the thermodynamic limitations on cloud deepening. Changes in domain-wide condensation and deposition contribute to change in condensate gain $\Delta G$ (SI Fig. 9). Condensation contributes most to the increases between the low and the standard aerosol scenario, while changes in condensation and deposition contribute about equally to $\Delta G$ between the standard and high aerosol scenario.

The condensate loss L, i.e. the integrated evaporation and sublimation rate, is more sensitive to the aerosol concentration than G and generally increases continuously with aerosol concentration (Fig. 7a). $\Delta L$ is larger for increasing than decreasing aerosol concentration relative to the standard aerosol scenario. This is also reflected in a lower precipitation efficiency PE (Fig. 7d). The changes are qualitatively similar in all ensemble member, although the absolute change in precipitation efficiency (and L) varies somewhat between ensemble members. The stronger decrease in PE from the standard to the high aerosol scenario

compared with the low and standard aerosol scenario is consistent with a higher lateral condensate transport to the stratiform region as hypothesised by Miltenberger et al. (2017). With further cloud deepening limited by the upper level stable layer, more condensate is transported into the stratiform area reducing the residence time of the condensate in the active convective core region. In contrast, cloud deepening is larger and changes in lateral condensate transport smaller when the low and standard aerosol scenario are compared. Therefore, the slower conversion of condensate to precipitation-seized hydrometeors in the

standard aerosol scenario can be partly balanced by a longer residence time in the convective core region. This hypothesis is discussed in more detail in Miltenberger et al. (2017).

The accumulated surface precipitation is shown in Fig. 7c. The surface precipitation is smaller in most simulations with the standard compared to the low aerosol scenario. Exceptions are the control simulation with a small increase in precipitation and ensemble members 6 and 8 with no change in accumulated surface precipitation. The latter have a relatively small decrease



of PE and a comparatively large $\Delta$G. Accordingly, the precipitation increase for these members is driven by the increase in G. For the other members, the change in PE dominates over changes in condensate production. If the aerosol concentration is enhanced beyond the standard scenario, the precipitation response in all ensemble members is dominated by PE. This analysis is supported by the position of the different simulation in the $\Delta$G-$\Delta$L diagram introduced in Miltenberger et al. (2017). All

points except those for the control, member 6 and 8 with low aerosol concentrations fall outside the shaded area (Fig. 10). The shaded area corresponds to the part of the phase-space dominated by changes in $\Delta$G (see Appendix of Miltenberger et al., 2017).

The decrease in accumulated precipitation is accompanied by a reduced mean precipitation rate with increasing aerosol concentrations (Fig. 8a). For most ensemble members the change is larger between the standard and high aerosol scenario than

between the standard and low aerosol scenario. Exceptions are ensemble members 3, 4, and 5, which also have the largest change in accumulated precipitation from low to standard aerosol concentrations. The percentiles of the precipitation distribution are compared in Fig. 8b. All percentiles up to and including the $75^{th}$ percentile show an increase with the aerosol concentration, while higher percentiles are generally smallest (largest) for the high (standard) aerosol scenario. This pattern is particularly evident for the $99^{th}$ percentile.

The condensed waterpath in the domain is a result of the condensate generation and the timescale of condensate conversion to precipitation. Parcel model considerations suggest a longer timescale for precipitation formation under enhanced aerosol concentrations. Therefore, an increase in the condensed waterpath is expected with increasing aerosol concentrations. The mean condensed waterpath in most ensemble members does increase with aerosol concentrations (Fig. 9a). This is the result of small decreases in the liquid water path (cloud and rain species) and a larger gain in the mass of the frozen hydrometeors (ice, snow,

and graupel species) (Fig. 9b and SI Fig. 10). This is consistent with a slower conversion of cloud droplets to rain drops and accordingly a larger mass transport across the $0\,^{\circ}$C level. The only ensemble member displaying a different pattern is ensemble member 9, for which the total condensed, the solid, and the liquid waterpath decreases from the standard to the high aerosol scenario. For this ensemble member, a strong reduction of the precipitation efficiency is observed. Also, the mean cloud top height and the fraction of cloud with cloud tops larger than $4.3\,\mathrm{km}$ decreases compared to the standard aerosol scenario. These

changes are consistent with a lower condensed water path, as reduced cloud top heights indicate a smaller vertical displacement of the air parcels and accordingly less condensate generation. The decrease in precipitation efficiency is likely linked to these changes as the longer timescale for conversion of cloud droplets to precipitation seized hydrometeors is not compensated by a longer residence time in the cloud due the reducing vertical extend of the clouds.

## 5.4 Radiation

The response of cloud radiative properties to changes in aerosol concentrations is climatologically important, but not well constrained mainly due to the impact of aerosols on cloud fraction and cloud lifetime. The reflected shortwave radiation is affected by the size and number of the hydrometeors close to the cloud top and the cloud fraction. The simulations with different aerosol scenarios suggest an increasing outgoing shortwave flux for higher aerosol concentrations (Fig. 11a). This change is consistent with the increasing CDNC and small impact of the aerosol scenario on the cloud fraction. The radiative signal presented here



is mainly due to CDNC changes and does not fully take into account potential changes in radiative properties of the ice phase species, as the effective diameter of the latter is diagnosed from the ice water content.

The outgoing longwave radiation is mainly influenced by the surface temperature, the cloud fraction, and the cloud top temperature. The mean outgoing longwave radiation shows only a small sensitivity to the aerosol scenario for all meteorological

ensemble members (Fig. 11b). The meteorological variability is mainly a result of changes in cloud fraction and cloud top height and to only a smaller degree the surface temperature. The small discernible trend of decreasing mean outgoing longwave radiation with increasing aerosol ($< 0.5\,\mathrm{W\,m^{-2}}$, standard to high aerosol scenario) is consistent with the small increase in mean cloud top height (Fig. 6d). Changes in cloud top height are partly compensated by a small negative trend in cloud fraction (Fig. 6c).

## 6  Ensemble with perturbed meteorological and aerosol initial conditions

In the previous two section, the response of cloud properties to perturbations of the aerosol or meteorological initial and boundary conditions has been discussed separately. The 10 meteorological ensemble members vary in the large-scale moisture convergence, thermodynamic profile, and wind shear. The variation in the large-scale moisture convergence is most important for the cloud field properties as its variations are reflected in the cloud fraction, cell number, condensate generation, and

accumulated precipitation. The mean cloud top height varies between ensemble members according to the different thermodynamic profiles. Aerosol-induced changes are similar for each meteorological ensemble member. An increase in aerosol number concentration translates to a larger CDNC, mean cell area, and outgoing shortwave radiation, while the cell number and precipitation efficiency decrease with increasing aerosol concentrations. The mean cloud top height, condensate generation, and outgoing longwave radiation display only very small changes in response to altered aerosol concentrations.

To detect aerosol-induced changes in cloud properties or precipitation formation with observational data-sets, it is important to separate changes resulting from slightly different meteorological conditions from changes resulting from different aerosol concentrations. This is necessitated by the co-variability of aerosol and meteorological conditions in the real atmosphere. The question of the relative importance of meteorological and the aerosol initial and boundary conditions for the cloud field structure and precipitation formation is also important for operational numerical weather prediction and the future design of

ensemble prediction systems. Here, we use the combined meteorological and aerosol initial condition ensemble, i.e. combining the discussion of the two previous section, to address the question of the relative importance of aerosol and meteorological variability for the COPE case. The discussion will focus on changes in the 10 hour mean properties of the cloud field between 9 UTC and 19 UTC. The mean value of the considered variable is displayed along with its spread from the meteorological ensemble members on the left side of Figs. 5-11 for each aerosol scenario (different colours). If instantaneous realisation of

the different (domain-averaged) variables are considered the variability is much larger as is illustrated by the box-plots on the left side of each plot.

The cloud droplet number concentration at either cloud base or cloud top is strongly influenced by the assumed aerosol scenario, but varies little between the different meteorological members. As a result, a clear aerosol signal remains present when





the meteorological variability is taken into account (Fig. 5). Although there is a stronger meteorology induced variability in the cell number and mean cell size (Fig. 6 a and b) and the predicted range of values overlap for different aerosol scenarios, the aerosol signal is clearly detectable in these variables. However, if the cloud fraction, the mean cloud top height or the distribution into cloud top height classes is considered the meteorological variability dominates and the difference between the

different aerosol scenarios is very small (Fig. 6 c and d, SI Fig. 8 a). Considering previous arguments on convective invigoration, it is interesting to note that the cloud top height varies only very little with aerosol scenario, but is sensitive to changes in meteorological conditions. No significant differences in outgoing longwave radiation exist between the aerosol scenarios consistent with the small aerosol-induced changes in cloud top height, cloud fraction, and surface temperature (Fig. 11b). For the outgoing shortwave radiation a stronger aerosol signal is retained above the meteorological variability due to the large

impact of aerosol concentrations on CDNC (Fig. 11a).

The precipitation formation is known to be strongly influenced by dynamics and microphysical processes. Miltenberger et al. (2017) used an analysis of the water budget to separate the contributions from cloud dynamics and microphysics to aerosol-induced changes. This methodology can be tested with the combined aerosol and meteorological ensemble discussed here. The condensate gain and condensation ratio vary strongly between different meteorological ensemble members, but show little

sensitivity to the aerosol scenario (Fig. 7 a, b). The small dependency of the condensate gain on the aerosol number concentration may be a result of using a saturation adjustment scheme for the condensation in our model. Previous studies using a prognostic supersaturation found the condensation rates to be dependent on the CDNC number concentration (e.g. Lebo et al., 2012; Lebo, 2014; Sheffield et al., 2015). However, due to the thermodynamic constraints on integrated condensation, we do not expect this will have a strong impact on the overall behaviour of the condensate gain. The condensate loss shows a some-

what larger sensitivity to the aerosol scenario, but is also dominated by the meteorological variability due to the close relation with the condensate gain (Fig. 7a). The precipitation efficiency discounts this co-variability and accordingly display an even smaller dependency on the meteorology and a larger dependency on the aerosol scenario (Fig. 7 d).

The aerosol-induced change in accumulated precipitation is the combined result of the changes in condensation ratio and precipitation efficiency. While the accumulated surface precipitation in most meteorological variables decreases with increasing

aerosol scenario, these differences are much smaller than the variability of accumulated surface precipitation between meteorological ensemble members (Fig. 7 c). The meteorological variability is due to large differences in the condensate gain, which is directly related to the variability in large-scale moisture convergence. The aerosol signal is much larger for an increase of the aerosol concentrations beyond the standard aerosol scenario due to a significantly larger change in the precipitation efficiency. The mean precipitation rate behaves qualitatively very similar to the accumulated precipitation (Fig. 8a). An increase

of the aerosol concentration decreases the occurrence frequency of low and medium precipitation rates (percentiles up to the 75[th] increase, Fig. 8b). High precipitation rates are most common in the simulations with the standard aerosol profile (99[th] percentile is largest). However, the meteorology induced variability in the percentiles is larger than these differences.

The increasing aerosol concentrations shift the distribution of ensemble-mean condensed water path to higher values, if all hydrometeor types are considered (Fig. 9 a). However, the liquid water path (condensate in the cloud and rain category) shows

little sensitivity to the aerosol scenario. This indicates an increasing frozen fraction with increasing aerosol concentrations



consistent with a longer timescale for precipitation formation. The aerosol-induced change in both variables are much smaller than the variability induced by different meteorological initial and boundary conditions.

## 7   Discussion and Conclusions

High-resolution ensemble simulations ($\Delta x = 250\,\mathrm{m}$) with perturbed aerosol and meteorological initial and boundary conditions were performed for convection forming along sea-breeze convergence zones over the southwestern peninsula of the UK. The relative importance of perturbations in meteorological (10 members) and aerosol initial conditions (3 for each member) for various cloud properties and precipitation formation is analysed over a forecast leadtime of $10 - 20\,\mathrm{h}$. The 10 different meteorological ensemble members develop similar meso-scale flow patterns with a sea-breeze convergence zone establishing over the centre of the peninsula. As a result of the different lateral boundary conditions, the large-scale boundary layer moisture convergence and the accumulated condensate gain vary by a factor 2 and the accumulated surface precipitation by a factor 2.5 between ensemble members. The average cloud fraction differs by up to 0.1 between the meteorological ensemble members. This meteorological variability is compared to changes in cloud properties induced by a factor 10 increase and decrease of aerosol number concentrations relative to the standard scenario. While the perturbations to the meteorological initial conditions reflect at best the uncertainty for the investigated case, changes in aerosol number concentration by a factor 100 are probably even larger than what could be expected for the climatological variability.

Changes in aerosol concentrations modify cloud field properties, e.g. cell number and size, cloud depth, cloud fraction, and the domain-wide condensate budget (condensate gain and loss, precipitation rate). The variability of cloud field properties across the ensemble is summarised in Fig. 12. Aerosol-induced changes are consistent across the ensemble suggesting that the physical mechanism discussed by Miltenberger et al. (2017) is robust against small changes in meteorological initial conditions. The possibility of discerning aerosol-induced differences in various cloud metrics relative to realistic meteorological variability is assessed in the following. First, the idealised situation where the meteorological initial conditions are identical for different aerosol perturbations is assessed by pairing ensemble members according to the meteorological initial conditions. For the paired ensemble members, a factor 10 increase or decrease in aerosol concentrations introduces statistically significant changes (at the $5\,\%$ level) in CDNC, cloud fraction, cell number and size, outgoing shortwave radiation, instantaneous and mean precipitation rates, and precipitation efficiency (table 1). Note that the statistical analysis is based on a very small sample, which affects the validity of several assumptions. However, since the indications agree with the physical analysis, we use the significance values as a helpful diagnostic for summarising the results. Aerosol-induced changes in accumulated precipitation are only significant for an increase of aerosol concentrations beyond the standard scenario. An analysis of the condensed water budget suggests that for a decrease in aerosol concentrations, a smaller condensation ratio is balanced by an increasing precipitation efficiency. In contrast, for higher aerosol concentrations than in the standard scenario, the precipitation response is dominated by a strong decrease in precipitation efficiency with little change in the condensation ratio due to the thermodynamic constraints on cloud top height.





Secondly, we can use the simulations to assess our ability to discern aerosol-cloud effects for the situation where meteorological initial and boundary conditions are similar but subject to observational uncertainty. This would represent a "perfect" observational campaign where the meteorological conditions each day are only but slightly different (convergence within a factor 2) and large perturbations to aerosol concentrations occur (factor $10 - 100$). This scenario is replicated by analysing

aerosol-induced changes in the full ensemble without pairing ensemble members according to meteorological initial conditions. For the un-paired ensemble, only aerosol-induced changes in CDNC, cell number and size, outgoing shortwave radiation, and precipitation efficiency are statistically significant (table 1). For some of these variables, the changes are significant only for a decrease or increase of aerosol number concentration relative to the standard scenario. For all other investigated variables (cloud fraction, cloud top height, condensation ratio, domain-average precipitation rate, condensed water path, and liquid water

path) the variability resulting from different meteorological initial and boundary conditions is equal or larger than the aerosol-induced signal.

The ensemble data can be used for a rough estimate of the number of observations that are required for retrieving a robust aerosol signal from observational data for sea-breeze convection. For this analysis we assume (i) the aerosol scenario and meteorology are independent, (ii) the ensemble is representative of the meteorological variability, (iii) the meteorological vari-

ability can be described by a Gaussian distribution, and (iv) observational data is perfect. While it is difficult to a priori estimate the impact of these assumptions on the analysis, we expect the analysis to provide a lower limit of the required number of observations required for the following reasons: In contrast to assumption (i), aerosol and meteorological conditions are likely to be correlated (e.g. Brenguier et al., 2003; Naud et al., 2016; Wood et al., 2017) reducing the observed section of the phase-space. Secondly, the meteorological variability in the ensemble simulations is not representative of the climatological variability of

meteorological conditions for sea-breeze convection over the the southwestern peninsula of the UK, which can be assumed to be much larger (Golding, 2005). Lastly, observational data will not be perfect due to measurement errors and spatial and temporal sampling issues (e.g. Schutgens et al., 2017). All these issues will likely increase the number of required samples compared to the values suggested by our analysis.

With the assumptions listed above, a Gaussian distributions representing the meteorological variability is defined for the low

and the high aerosol scenario. For each variable, the mean value across all ensemble members with the same aerosol scenario is used as the mean of the Gaussian distribution. The standard deviation $\sigma$ of the Gaussian distribution is defined by assuming the value range (minimum and maximum) across the ensemble members equals $4\,\sigma$ ($3\,\sigma$ and $5\,\sigma$ are tested as well). Then $10^4$ realisations with n samples are drawn from the Gaussian distributions for the low and high aerosol scenario separately. The number of samples n can be interpreted as the number of times the same day is observed, as the statistical analysis presented here uses

either daily average or accumulated variables. However, it may be possible to interpret the necessary number of observations also as number of individual observations, e.g. from satellite overpasses, if subsequent observations are not auto-correlated, i.e. are from different cloud lifecycles. If snapshots are used, it may be necessary to take into the possibly different life cycle stages of the observed cloud field (e.g. Luo et al., 2009; Witte et al., 2014). However, given the limited number of ensemble members in our analysis, an assessment of this effect is beyond the scope of our study. For each realisation, we test the hypothesis that the

low and high aerosol scenario are not equal with a two-sided t-test. The resulting distribution of p-values gives the probability





that a significant aerosol-signal can be retrieved from a sample of n observations with low and high aerosol conditions each (SI Fig. 11). The number of days required to have a 95 % change to observe a significant aerosol-induced change in various cloud properties is listed in table 2. The required number of observations as given in the table only gives an approximate indication, while the exact numbers is sensitive to the assumptions made regarding the presentation of the meteorological variability in the

ensemble, e.g. whether the ensemble spread corresponds to $3\sigma$, $4\sigma$, or $5\sigma$. It is important to note that the statistical analysis has the strong caveat of being based on a rather small ensemble. To obtain robust statistics a much larger ensemble with several 100 ensemble members would be required, which is currently beyond the computational resources available. However, we think the analysis provided here gives some general indication of the scale of observations required as the statistics confirm the impressions gained from the physical analysis of the ensemble members. Our analysis indicates that a small sample $n \leq 10$

is sufficient for variables such as the CDNC and outgoing shortwave radiation, while a large sample often exceeding 100 is required for variables such as cloud fraction, cloud top height, or accumulated precipitation. The number of samples required depends on the location in the aerosol space, i.e. different low and standard or high and standard aerosol scenario are considered. In general, more observations are required for an increase of aerosol number concentrations above the standard scenario, which is related to the thermodynamic constraints on aerosol-induced changes discussed in Miltenberger et al. (2017).

While the meteorological ensemble allows to put the aerosol-induced changes in cloud properties into the context of changes related to meteorological variability, the considered changes in meteorology are fairly small (section 4). Even if the represented meteorological variability is assumed to be representative of all possible meteorological conditions on the investigated day, they do not cover the full range of meteorological conditions that could occur for convection along sea-breeze convergence zones. However, even this very conservative estimate on the meteorological variability is for many variables on the same order of

magnitude or larger than the aerosol-induced changes. We expect that the number of samples required to retrieve a statistically robust aerosol-induce change would increase, if the climatological variability of the meteorological conditions is considered. The results presented in this paper certainly only pertain to the specific cloud type investigated and the relative magnitude of aerosol and meteorological related changes in cloud properties may be different for other cloud types. This will be investigated in future studies. In addition to the results presented here, some previous studies have highlighted the importance of considering

the intrinsic predictability of investigated cases before drawing conclusions about the significance of aerosol-induced changes in cloud properties (Grabowski et al., 1999; Khairoutdinov and Randall, 2003; Zeng et al., 2008; Morrison and Grabowski, 2011; Morrison, 2012). These studies used prescribed large-scale conditions and applied random perturbations to thermodynamic fields throughout the simulations. The present study complements their analysis by considering the impact of changes in the large-scale conditions, which are small compared to observational uncertainties and much smaller than the expected

variability in meteorological categories used to retrieve aerosol signals from general circulation models (e.g. Bony et al., 2004; Zhang et al., 2016). Consistent with previous studies, we find that the aerosol signals in variables closely related to aerosol concentrations are easier to retrieve than for variables that are linked to aerosol concentrations by a series of complex processes. From the limited number of studies available, the set of variables in either category appears to vary for different cloud types and geographic location. However, our and previous studies all suggest that aerosol-induced change in surface precipitation are

very difficult to retrieve reliably (e.g. Khairoutdinov and Randall, 2003; Morrison and Grabowski, 2011).



The evidence presented in the to-date very limited number of studies considering the relative impact of meteorological and aerosol conditions on cloud properties suggest that it is crucial to carefully consider intrinsic predictability, meteorological conditions, and co-variability between aerosol and meteorological conditions in modelling and observational studies of aerosol indirect effects. While these aspects have been highlighted by Stevens and Feingold (2009) and Feingold et al. (2016), only

5    a few modelling studies have investigated these aspects and there is a clear need for future studies extending the analysis to other cloud types and meteorological scenarios. An improved knowledge and quantification of these aspects is mandatory for progress in our understanding of aerosol-induced changes in cloud properties and for retrieving observational evidence thereof.

*Data availability.*    Model data is stored on the tape archive provided by JASMIN (http://www.jasmin. ac.uk/) service. Data access to Met Office data via JASMIN is described at http://www.ceda.ac.uk/blog/access-to-the-met-office-mass-archive-on-jasmin-goes-live/.

10    *Author contributions.*    All authors contributed to the development of the concepts and ideas presented in this paper. B. J. Shipway developed the CASIM microphysics code. A. A. Hill, J. M. Wilkinson, P. R. Field and A. K. Miltenberger contributed to the further development of the CASIM code. A. K. Miltenberger and P. R. Field helped set up the model runs. A. K. Miltenberger performed the model simulations and model analysis, and wrote the majority of the manuscript, along with input and comments from all co-authors.

*Competing interests.*    The authors declare that they have no conflict of interest.

15    *Acknowledgements.*    We thank the COPE research team for collecting observational data and Alan Blyth for very useful discussion input. We acknowledge use of the Monsoon/NEXCS system, a collaborative facility supplied under the Joint Weather and Climate Research Programme, a strategic partnership between the Met Office and the Natural Environment Research Council. Further we acknowledge JASMIN storage facilities (doi : 10.1109/BigData.2013.6691556) as well as FAAM, CEDA, BADC, and the Met Office for providing data. The University of Leeds is acknowledged for providing funds for this study.



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





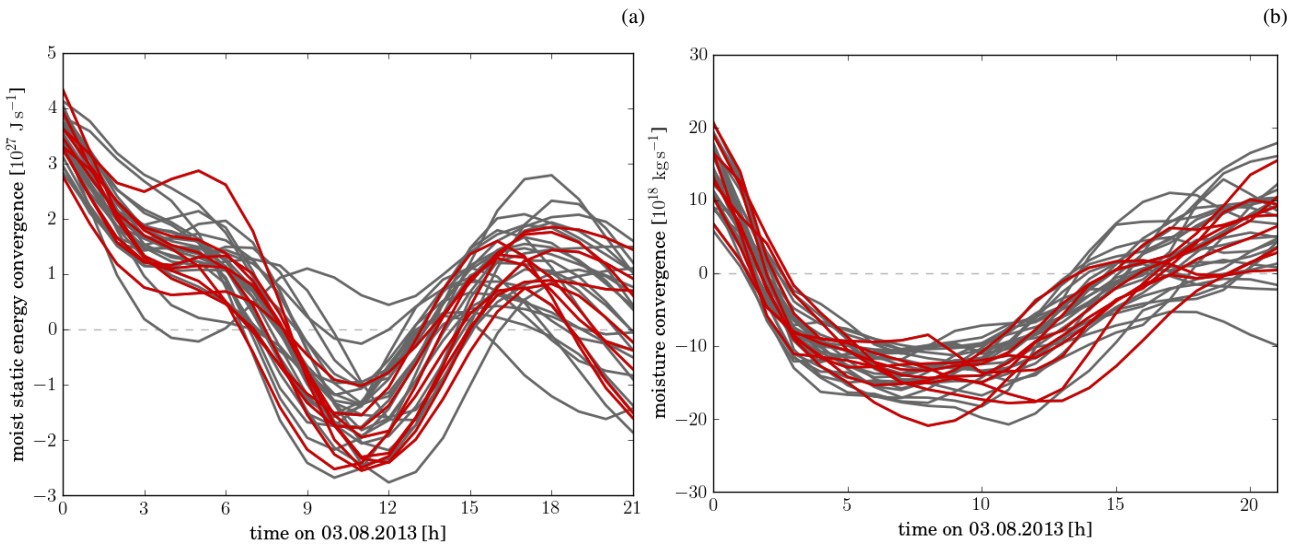

**Figure 1.** Convergence of moist static energy (a) and moisture (b) across the 1 km domain from the global ensemble. The grey lines show all 33 ensemble members in the global ensemble and the red lines the 9 members selected for the regional ensemble simulations.





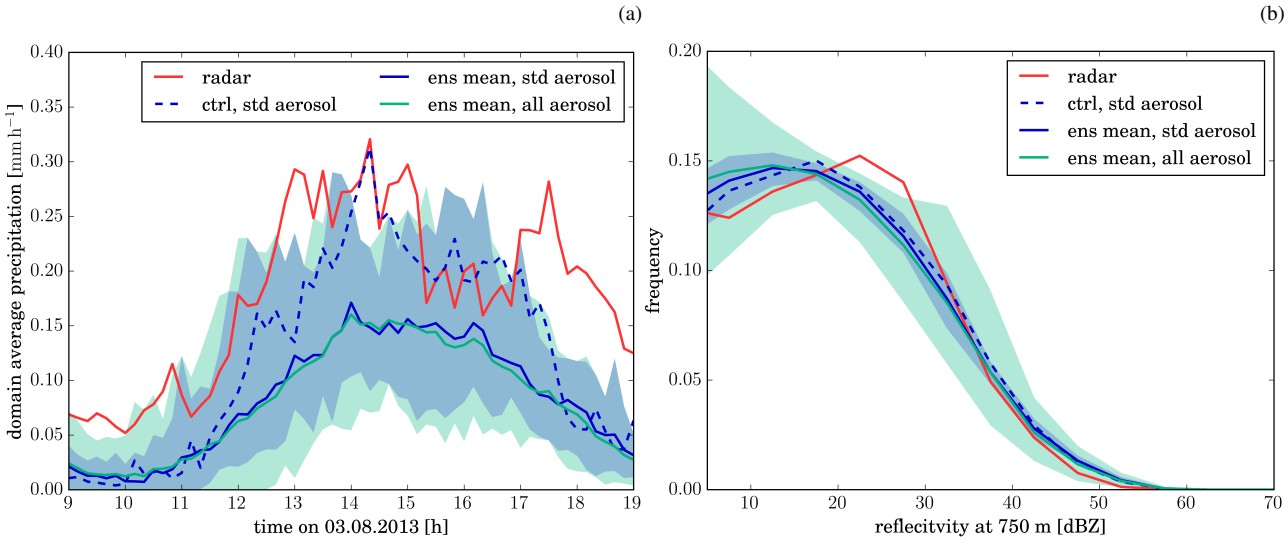

**Figure 2.** Comparison of domain-mean surface precipitation (a) and radar reflectivity at 750 m (b) from model simulations and radar observations (red line). Values from the control simulation with the standard aerosol profile are shown by the dark blue dashed line. The mean (envelope) of all ensemble members using the standard aerosol profiles is shown by the dark blue solid line (shading) and those of all ensemble members irrespective of the used aerosol profile by the solid cyan line (shading).



**Figure 3.** Wind speed (a) and wind direction (b) at the upstream (western) domain boundary at various altitudes (colours). The temperature difference between land and sea is shown in (c) for different times (colours). The low-level convergence over the peninsula is displayed in (d).





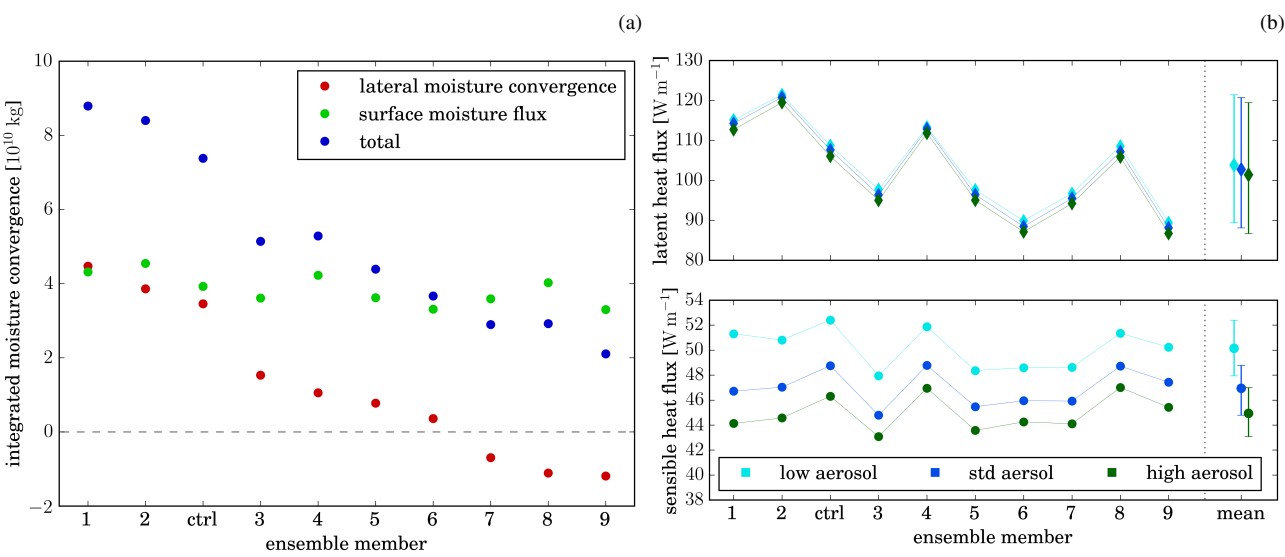

**Figure 4.** Time integrated incoming (blue) and outgoing (red) moisture flux (a) at the domain boundaries for each ensemble member. Panel (b) shows the time integrated moisture convergence (green) as well as its separation into positive (convergence, blue) and negative contributions (divergence, red).





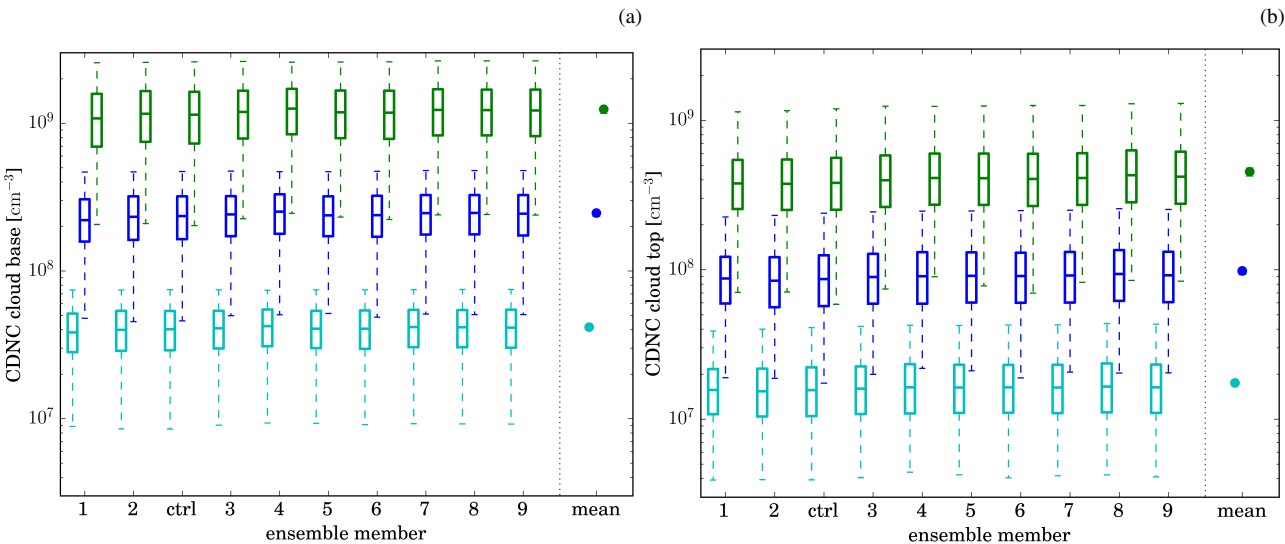

**Figure 5.** Cloud droplet number concentration (CDNC) at cloud base (a) and cloud top (b) for different ensemble members (abscissa) using different aerosol profiles (colours). CDNC at cloud base (top) is computed as the average CDNC within $500$ above (below) the lowest (highest) point in each grid column that has a cloud or ice mass mixing ratio larger than $1\,\mathrm{mg\,kg^{-1}}$. The horizontal line inside the boxes indicates the mean CDNC, the upper and lower edges the $25^{\mathrm{th}}$ and $75^{\mathrm{th}}$ percentile, respectively, and the whiskers the $1^{\mathrm{st}}$ and $99^{\mathrm{th}}$ percentile. The statistics are computed over all qualifying grid points in the domain between $10\,\mathrm{UTC}$ and $18\,\mathrm{UTC}$ and therefore reflect the spatial and temporal variability of CDNC. The last column provides the distribution of the ensemble means, with the dot representing the average of the ensemble means and the bars the spread of the ensemble means.



**Figure 6.** Cell number (a), mean cell size (b), cloud fraction (c) and mean cloud top height (d). Cells are defined as continuous areas of column maximum radar reflectivity exceeding 25 dBZ. Cloud fraction is fraction of the domain, for which the condensed water path is larger than $1\,\mathrm{g\,m^{-2}}$. Cloud top height is the height of the highest vertical level in each grid column with a condensate mass mixing ratio larger than $1\,\mathrm{mg\,kg^{-1}}$. The horizontal line inside the boxes indicates the time mean value, the upper and lower edges the $25^{\mathrm{th}}$ and $75^{\mathrm{th}}$ percentile, respectively, and the whiskers the $1^{\mathrm{st}}$ and $99^{\mathrm{th}}$ percentile. These statistics reflect the temporal variability of the considered variables. The last column in each panels provides the distribution of the ensemble means, with the dot representing the average of the ensemble means and the bars the spread of the ensemble means.



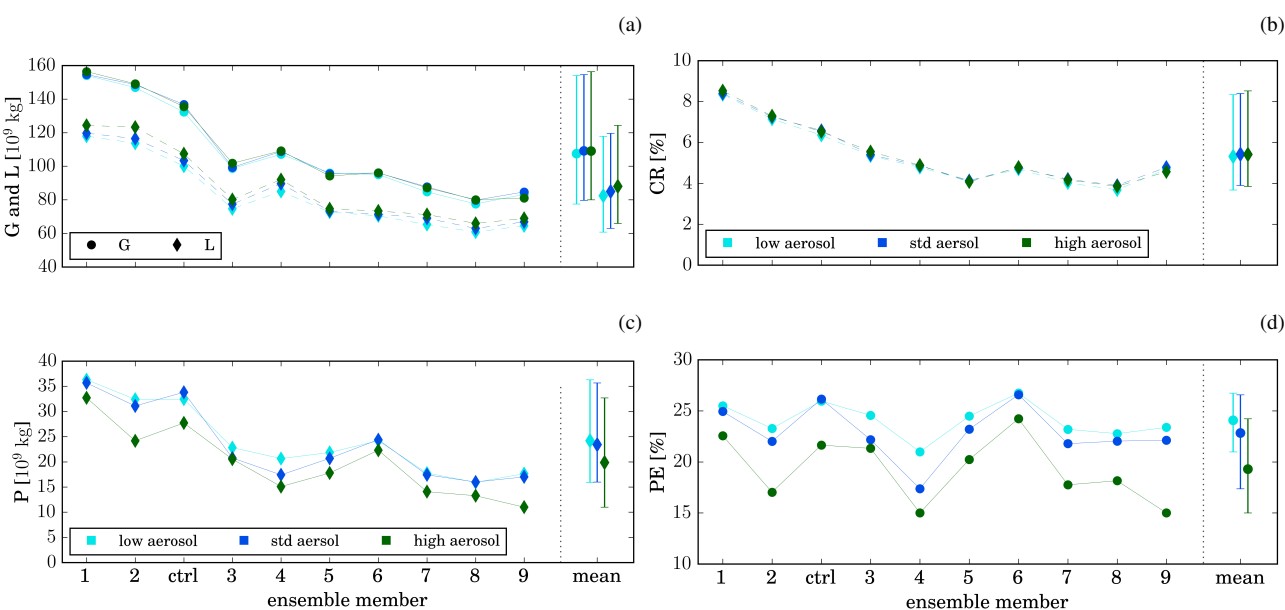

**Figure 7.** Accumulated condensate generation and loss (a) and surface precipitation (c) for the different ensemble members and aerosol profiles between $9 - 20$ UTC. Panel (b) shows the condensation ratio and (d) the precipitation efficiency for the same time period. The last column in each panels provides the distribution of the ensemble means, with the dot representing the average of the ensemble means and the bars the spread of the ensemble means.



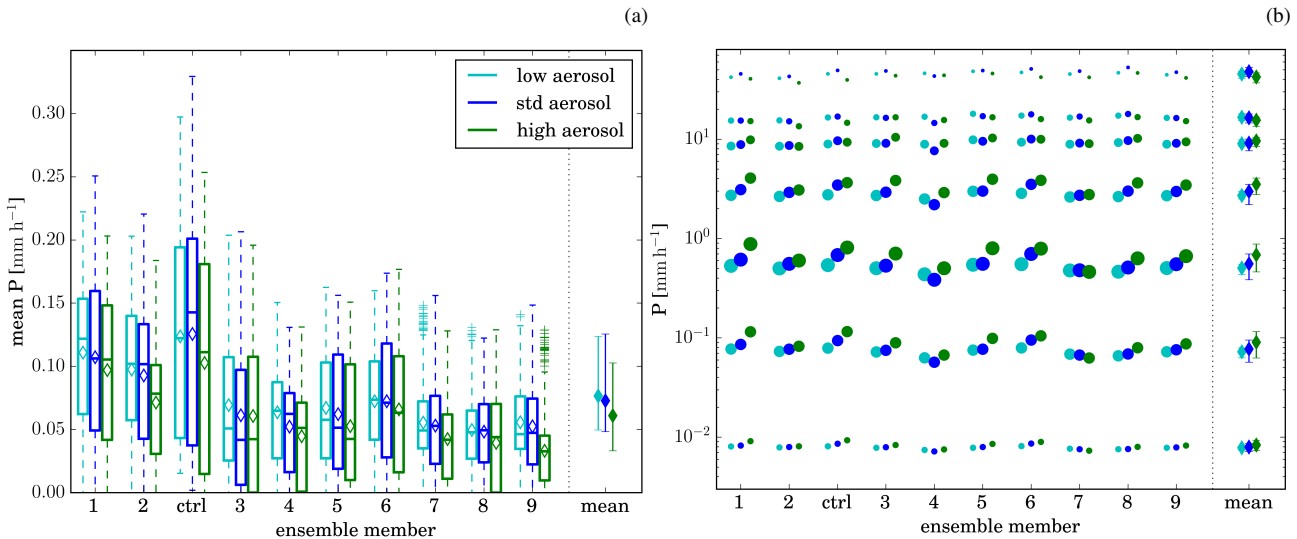

**Figure 8.** Domain-average precipitation rate (a) and the percentile of the instantaneous rain rate (b). The $99^{th}$, $95^{th}$, $90^{th}$, $75^{th}$, $50^{th}$, $25^{th}$ and $5^{th}$ percentiles are shown from top to bottom. The horizontal line inside the boxes indicates the time mean value, the upper and lower edges the $25^{th}$ and $75^{th}$ percentile, respectively, and the whiskers the $1^{st}$ and $99^{th}$ percentile. These statistics reflect the temporal variability of the considered variables. The last column in each panels provides the distribution of the ensemble means, with the dot representing the average of the ensemble means and the bars the spread of the ensemble means.





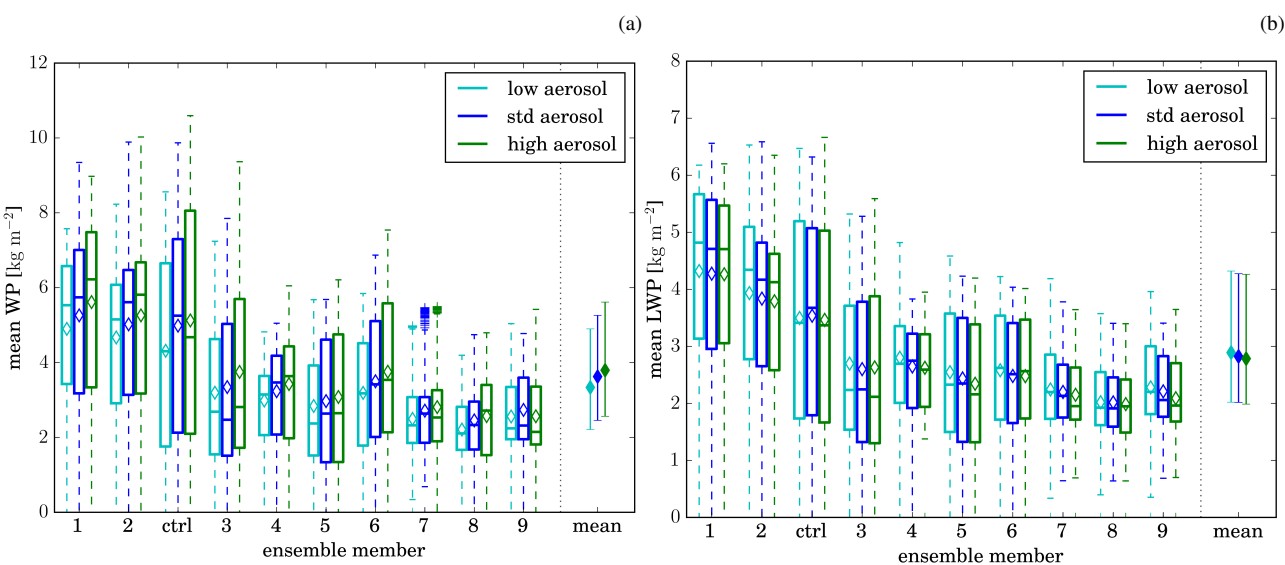

**Figure 9.** Condensed water path (including all liquid and solid hydrometeors) (a) and liquid water path (including cloud and rain) (b). The horizontal line inside the boxes indicates the time mean value, the upper and lower edges the $25^{\text{th}}$ and $75^{\text{th}}$ percentile, respectively, and the whiskers the $1^{\text{st}}$ and $99^{\text{th}}$ percentile. These statistics reflect the temporal variability of the considered variables. The last column in each panels provides the distribution of the ensemble means, with the dot representing the average of the ensemble means and the bars the spread of the ensemble means.



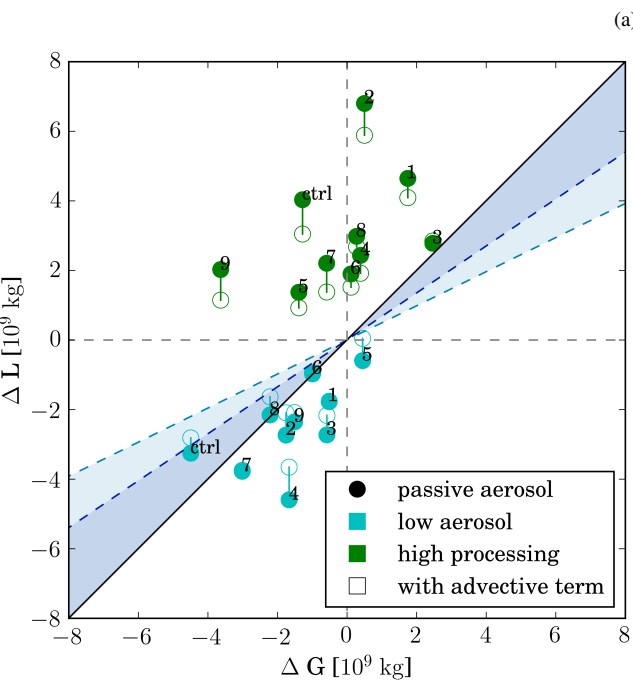

**Figure 10.** $\Delta$G in relation to $\Delta$L for ensemble members paired according to the meteorological initial conditions. $\Delta$G and $\Delta$L are computed for simulations with the high (green symbols) and low (cyan symbols) aerosol profile relative to the simulations with the standard aerosol profile. The black symbols indicate $\Delta$G and $\Delta$L as results from using mean G and L terms of all ensemble members with a specific aerosol profile (low: downward pointing triangle; high: upward pointing triangle).





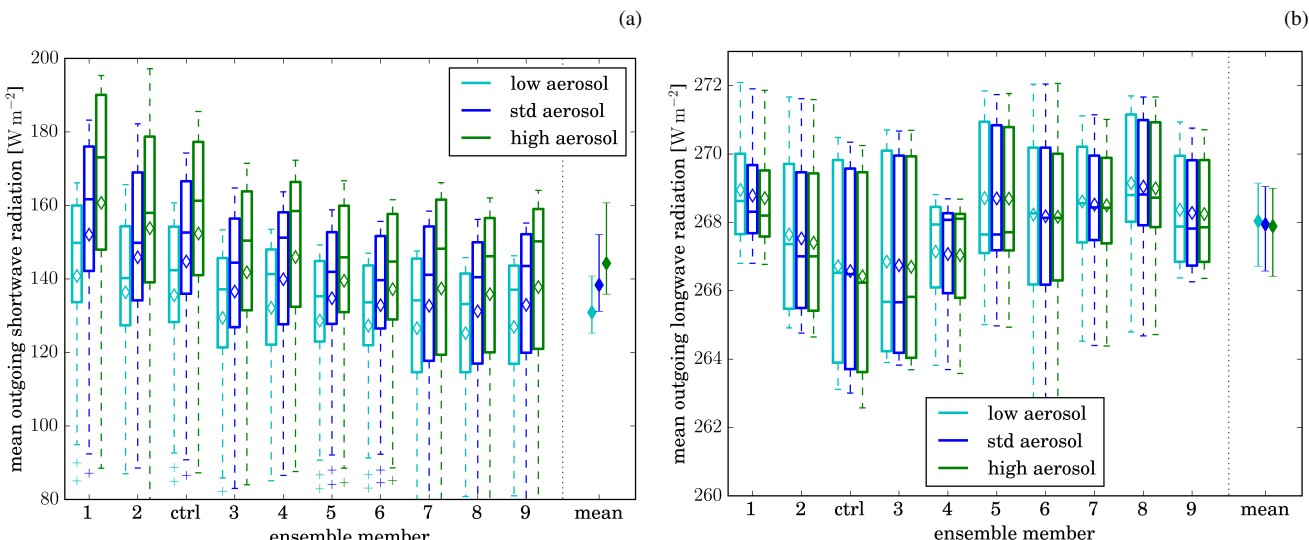

**Figure 11.** Outgoing shortwave (a) and longwave (b) ratio at the tropopause. The horizontal line inside the boxes indicates the time mean value, the upper and lower edges the $25^{th}$ and $75^{th}$ percentile, respectively, and the whiskers the $1^{st}$ and $99^{th}$ percentile. These statistics reflect the temporal variability of the considered variables. The last column in each panels provides the distribution of the ensemble means, with the dot representing the average of the ensemble means and the bars the spread of the ensemble means.





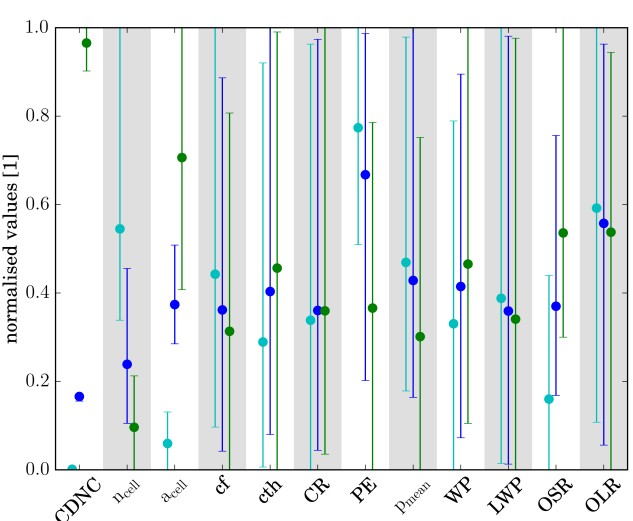

**Figure 12.** Summary of variability in time average $(10 - 19\,\mathrm{UTC})$ cloud properties induced by variations in meteorological initial conditions (bars) and aerosol initial conditions (colours). Each variable has been normalised such that the minimum and maximum values in the entire ensemble (aerosol and meteorology) map to the value range $[0, 1]$.



**Table 1.** p-values from two-sided t-test with the null hypothesis of no change in the variable (columns) between two aerosol scenarios (rows) for all ensemble members. The results for ensemble members paired according to meteorological conditions and un-paired members are provided.

| | low - standard | | standard - high | | low - high | |
|---|---|---|---|---|---|---|
| | paired | unpaired | paired | unpaired | paired | unpaired |
| cloud fraction | **0.00154** | 0.643 | **0.00310** | 0.737 | **0.00190** | 0.431 |
| cell number | **8.99e-6** | **0.00178** | **1.49e-5** | **0.00591** | **7.68e-6** | **8.33e-5** |
| cell area. | **2.09e-7** | **7.11e-8** | 0.379 | 0.464 | 0.241 | 0.266 |
| OSR | **6.80e-7** | **0.0154** | **9.27e-7** | 0.113 | **7.63e-7** | **0.000799** |
| OLR | **8.33e-5** | 0.817 | **0.00576** | 0.894 | **0.000373** | 0.717 |
| cloud top height | **0.000243** | 0.325 | 0.549 | 0.678 | 0.104 | 0.204 |
| deep cloud fraction | **2.61e-6** | 0.222 | 0.465 | 0.914 | **2.63e-5** | 0.263 |
| CDNC cloud base | **8.31e-16** | **1.23e-15** | **7.52e-15** | **1.09e-14** | **5.26e-15** | **6.15e-15** |
| mean precipitation rate | **0.0123** | 0.748 | **0.000555** | 0.313 | **3.16e-5** | 0.174 |
| P | **0.0144** | 0.701 | **0.000372** | 0.248 | **2.84e-5** | 0.120 |
| G | **0.00501** | 0.896 | 0.803 | 0.991 | **0.0363** | 0.906 |
| L | **1.15e-4** | 0.794 | **0.000190** | 0.753 | **2.08e-5** | 0.571 |
| PE | **5.78e-3** | 0.273 | **1.31e-4** | **0.0145** | **1.32e-5** | **9.18e-4** |
| CR | **0.00140** | 0.874 | 0.878 | 0.994 | **0.0164** | 0.8823 |
| condensed WP | **0.000258** | 0.5323 | **0.00748** | 0.730 | **0.000176** | 0.342 |
| frozen WP | **1.13e-5** | **0.0159** | **0.000341** | 0.222 | **9.17e-6** | **0.00192** |
| liquid WP | **0.00450** | 0.848 | **0.0152** | 0.905 | **0.000477** | 0.756 |
| cloud WP | **2.34e-6** | 0.144 | **6.99e-6** | 0.396 | **2.81e-6** | **0.031** |



**Table 2.** Number of observation days to obtain a statistically significant (at the 5 % level) aerosol signal in 95 % of all cases. The main value assumes the spread of the meteorological ensemble members equals $4\,\sigma$, while the values in brackets use $5\,\sigma$ and $3\,\sigma$.

| Variable | aerosol within factor 100 (low to high scenario) | aerosol factor 10 lower (low to standard scenario) | aerosol factor 10 higher (standard to high scenario) |
|---|---|---|---|
| CDNC | $< 10$ | $< 10$ | $< 10$ |
| | $(< 10, < 10)$ | $(< 10, < 10)$ | $(< 10, < 10)$ |
| cloud fraction | 90 | 250 | 480 |
| | (50, 140) | (130, 340) | (320, 860) |
| cloud top height | 60 | 110 | 540 |
| | (40, 100) | (70, 190) | (350, 950) |
| outgoing SW | $< 10$ | 20 | 30 |
| | $(< 10, < 10)$ | $(< 10, 20)$ | (20, 50) |
| outgoing LW | 460 | 1110 | 3350 |
| | (290, 810) | (710, 1960) | (2180, 6000) |
| ac. precipitation | 30 | 420 | 60 |
| | (20, 60) | (290, 790) | (40, 90) |