# Peer review of "Aerosol-cloud interactions in mixed-phase convective clouds. Part 2: Meteorological ensemble."

_Atmospheric Chemistry and Physics, 2018_

## Referee Comment (RC1) · Anonymous Referee #1 · 21 Mar 2018

**Review of "Aerosol-cloud interactions in mixed-phase convective clouds. Part 2: Meteorological ensemble" by Miltenberger et al.**

The authors perform ensemble simulations of a case of deep convection forming along a sea-breeze convergence line in the southwestern UK. Detailed analysis of the case is first presented in the accompanying paper Part 1. Using an ensemble of simulations, the authors perform concurrent investigations into the variability in simulated cloud properties due to meteorology and the variability due to aerosol scenario. A total ensemble of 30 members is used, comprising 10 meteorological members (produced by downscaling a global ensemble with perturbed initial conditions) and then for each meteorological member employing 3 aerosol scenarios (a standard case close to that observed in accompanying Part 1, and a low and high case with respect to the standard case). When ensemble members are paired according to the meteorological initial and boundary conditions, aerosol-induced changes to the cloud properties are found to be consistent across the ensemble. The authors find statistically significant aerosol-induced changes to the cloud droplet number concentration, cloud fraction, convective cell number and size, outgoing shortwave radiation and precipitation efficiency. The authors conclude that for most cloud properties a large number of ensemble members (order 100 or more) of meteorological conditions is required to detect a robust aerosol effect. Only for impacts on cloud droplet number concentration and shortwave radiation are small sample sizes sufficient.

The manuscript generally presents and explains the results clearly, and I find it to be a very useful scientific contribution to the literature, where case studies rarely employ ensemble methods (especially when high-resolution cloud-resolving models are used). However, there are many typos and language errors scattered throughout the manuscript which must be addressed before publication. Further, I find the figure ordering and the placement of a couple of key figures in the supplementary information to be somewhat detrimental to the reader's comprehension, and I would recommend giving this careful attention before publication.

I have provided detailed comments below. I recommend publication in ACP subject to major revisions.

**General comments:**

1. The manuscript contains many typos and errors that the team of authors should really have addressed together through proof-reading before submission for review. I have listed these in my specific comments.

2. I believe there are some inconsistencies in the references to ensemble members. I have noted these in my comments on Section 4.2.

3. **Figure ordering:** I found the order of figures somewhat counterintuitive and hard to follow.

I had to lay my printed copy of the figures and the supplementary figures out next to each other in order to follow the arguments made in the text. Whilst I appreciate that there are

already a lot of figures in the paper, I would suggest moving figure S8 to the main paper if possible.

Further, the figure ordering in the main paper is not logical. I appreciate that it is difficult to optimally order figures when investigating concurrent sensitivities, but I would recommend placing the order of current Figures 7,6,5 as such. On P10 L4, the reference to Fig 7a, I had to jump ahead several Figures in order to see this. On P10 L34 I had to jump back again to Fig. 5, which is referenced for the first time after Fig.s 7 and 6 are discussed in detail. Why put Fig 5 in its current location? You would make it much easier for the reader if it appeared after 6 and 7.

Page 11, 1$^{st}$ paragraph: again, you refer to Fig 9 and then immediately after to Fig 11, and Fig 10 is not even mentioned until page 13.

**4. General comments on figures:**
Many of the figures have lines joining the points representing each ensemble member. This is misleading, as the abscissa on these figures show ensemble members (a discrete dataset) and not continuous data. I recommend removing these lines.

Figure SI 9 – I tried very hard to understand this Figure, but it many things in it don't make sense to me. See notes under my comments referring to individual figures.

**5. Section 2:**
Stochastic physics – are stochastic physics used in the regional model as well as the global model? Are stochastic physics used in the full set of ensemble runs? (Are you using stochastic physics as well as perturbed initial conditions?) What kind of stochastic physics are used? Which schemes and which parameters? Etc. This needs a little more explanation if you are discussing a study which aims to capture meteorological variability.

**6. Section 3.1:**
P6 L13 – Was the model microphysics output passed offline through the same radar algorithm as the Radarnet data? If not, could part of the difference be because the online UM dBZ calculation is different from the dBZ calculation in the Radarnet algorithm?

**7. Section 4.1:**
P9 L5-6: "These members have a higher cloud fraction" – do you know why this is the case for these members?
P9 L17-18: Is this also related to the cloudiness (higher cloud fraction in these members)?

**8. Section 4.2:**
P10 L11-12: Members 4 and 7 have a particularly large surface sensible heat flux – can you explain why? It doesn't seem like they stand out in terms of cloudiness (Fig. 6).

P10 L22: largest (smallest) values for ensemble members 8(4) – I find it hard to see by eye on this Figure, but doesn't this actually apply to members 9(5) not 8(4)?

P10 L22-23: Really? I find this hard to see (Fig 6d vs Fig SI 7c)

P10 L25-26: ensemble members 1,2,5,8 have a relatively large fraction of deep clouds – I don't see this. What about e.g. member 6 (Fig SI 8a)?

P10 L29: changes in condensate generation, i.e. air mass lifting – have you looked at the dynamical convergence to see if this is the case?

P10 L32:
- "member 8 has a relatively large PE" – I disagree with this. Many others have a greater PE, e.g. 1, ctrl, 6 (Fig. 7)
- "and the largest fraction of clouds with tops above 4.3 km" – I also disagree with this. The largest fraction of clouds with tops above 4.3 km is seen in member 6 (Fig SI 8a).

I think in this sentence perhaps the authors mean to refer to member #6, not member #8? Then I agree with the statements made in the sentence.

P11 L1-4: this final section is not particularly well-explained and no relevance is given. Can you say anything about the processes and impact or importance?

P11 L3-4: Largest (smallest) values occur for ensemble members 2(7) – I only just agree with this. Do you mean member 1 not member 2 for the largest SW radiation values and largest CF?

**Specific comments and typos:**
P1 L16: "consider" -> "considered"
P2 L3: "climate system"
P2 L4: "The main issues…"
P2 L5: "… on model grid scales several orders of magnitude larger, and the…"
P2 L6: "In the last few decades" / "In recent decades"
P2 L6-7: "the modification of cloud properties has been studied in particular"
P2 L7-8: "… and the relation between particle number concentrations and radiation" – this whole sentence feels quite clumsy.
P2 L13: "necessitated by" -> "necessary because of"
P2 L15: "changes to simulated for individual clouds" – simulated what?
P2 L17: You could also include a reference here to the 2012 paper by Seoung Soo Lee where placing an aerosol perturbation in the mesoscale domain of a simulation led to intensification of convection within an MCS but suppressed precipitation in the larger-scale domain. (Reference provided at the end of this set of comments)
P2 L29: Southern Great Plains
P3 L6: What do the authors mean by "cloud-induced changes to large-scale forcing"? Does this refer to large-scale circulation and / or synoptic forcing, or something else?
P3 L8: "has also" -> "also has"
P3 L10: relay -> rely
P3 L11: rises -> raises
P3 L12: datasets
P3 L12: has recently been demonstrated

P3 L14: Southern Great Plains

P3 L23: in future forecasting systems

P3 L28: 30th

P3 L34: baseline

P4 L2: a precipitation -> precipitation

P4 L4: the observed aerosol

P4 L7: convective invigoration hypothesis needs a description and / or citation

P4 L10: investigate whether the

P4 L17: Section number missing. (should this be Section 5?)

P4 L23-24: The way this sentence is written doesn't quite make sense.

P4 L25: do you mean "9 members are selected from", not "selected for"?

P4 L32: mesoscale

P5 L3-4: repetition of "current study";  you could just say "to our main conclusions".

P5 L6: In addition to (delete comma)

P5 L11: h a grid -> horizontal grid ?

P5 L32: datasets

P6 L3: peninsula (remove capital P)

P6 L16: have also reported

P6 L15: underdispersive over longer

P6 L31: smaller if (delete comma)

P7 L21, 23, 26: dewpoint

P7 L31: ensemble members

P9 L20: similar, with a well-mixed

P9 L25: temperate -> temperature

P10 L3,8: mesoscale

P10 L4: "G is very well correlated" – have you actually correlated this (or can you)?

P10 L4-5: Figures 4a and 7a are difficult to compare as they are on different pages

P10 L14: convergence

P10 L18: areal

P10 L26: an about 20% -> about a 20%

P10 L 10-31: refer to Fig SI 8a

P11 L3: largest (smallest) cloud fraction – please refer to Fig 6c.

P11 L4: distribution of cloud top heights (Fig. 11b) – you also need to refer to the Fig. showing CTH.

P 11 L8: "low", "high": open quotations are the wrong way wrong (LaTeX `` not "?)

P 11 L9: "which have a factor of 10 lower and higher aerosol number concentrations, respectively, than the standard profile"

P11 L10: altitudes

P11 L10: The mean and effective radius – mean what? Mean radius and effective radius?

P11 L12: the first section of this study?

P11 L13: ensemble members

P11 L17; Figure 5 should be moved, as discussed in the major comments

P11 L21: "suggest only minor changes in the cloud-base vertical velocity distribution" – can you plot this distribution? Doesn't this contradict the previous statement made about convergence?

P11 L24: "the number of cells decreases with increasing background aerosol concentration, but the cell area increases" – this is interesting! Can you explain why this happens?

P11 L30: cloud top height increases

P12 L2: "ensemble members 1,2,7,8, and 9" (missing space between "and9")

P12 L2: "ensemble members 1,2,7,8, and 9" – this is also true for member 4

P12 L3: "does not increase further (members 1 and 2)" – doesn't member 2 increase?

P12 L5: higher than 4.3 km shows only

P12 L7: aerosol scenario is likely (remove comma)

P12 L8: "maximum" (open quotation incorrect way round)

P12 L12: "only a small change"

P12 L14: "-4 – 2.5%" -  this notation is confusing. Do you mean -4% to -2.5%, or -4% to +2.5%?

P12 L18: I do not understand Figure SI9.

P12 L21: Can you plot delta G and delta L instead of G, L?

P12 L29: "seized" -> "sized"

P 12 L32: simulations in the standard

P12 L34: in Figure 7c in my printed copy, member 7 also looks like it has no change

P12 L34: "The latter have a relatively small decrease of PE and comparatively large delta G" – but member 1 also has a decrease in PE and delta G, but a decrease in precip in standard vs low scenarios, and is outside the shading in Figure 10.

P13 L1: "comparatively large delta G" – this is hard to see from Fig 7a. Can you plot delta G and delta L instead of G and L?

P13 L10: "Exceptions are ensemble members 3,4 and 5…" – you should point out that the behavior in each of these members is different from each other. For the (a) low to standard and (b) standard to high aerosol scenarios, member 3 has an (a) decrease and (b) increase, member 4 has an (a) decrease and (b) decrease, and member 5 has an (a) increase and (b) decrease.

P14 L11: two sections

P14 L20: datasets

P14 L29: realisations

P15 L4: distribution in different cloud top height classes

P 15 L11: Precipitation formation is known…

P1 L21: accordingly displays

P15 L29: very similarly to

P15 L34-35: "the liquid water path (…) shows little sensitivity to the aerosol scenario" – actually, there is a decrease in LWP (Fig 9b) which is not that much weaker than the increase in CWP (Fig 9a) – this indicates even more strongly than you currently state that the FWP must increase!

P16 L9: mesoscale

P17 L2: "perfect" (open quotations incorrect way round)

P17 L3: only slightly different

P18 L4: exact number is

P18 L7: several 100 ensemble members -> several hundreds of ensemble members

P18 L11: Why is low-high so different from low-standard and standard-high?

P18 L13: Accumulated precip stands out here – are you able to explaim why? (It's the only one that needs fewer observations for an increase of number concentration above the standard scenario).

P18 L14: "the thermodynamic constraints on aerosol-induces changes…" – constraints for this particular case, or general constraints?

P18 L15: allows us to put the aerosol-induced changes / allows the aerosol-induced changes to be put

**Comments on individual figures:**

Fig 4:
- Don't join these points with lines

Fig. 5:
- Can you comment on why these are so invariant?
- What do the colours represent?
- I think Figure 5 should appear AFTER Figure 6 (given the ordering of discussion in the manuscript)

Fig. 6:
- "cloud fraction is the fraction of the domain for which" (add "the", remove comma)

Fig 7:
- Fig. 7a would be clearer if you plotted delta G and delta L instead of absolute values
- Don't join these points with lines
- "the last column in each panel"
- Caption: "means" – do you mean ensemble mean, or ensemble means?

Fig 10:
- What do the open versus the filled circles represent?
- Legend: should "high processing" be "high aerosol"?
- Caption: "black symbols" – I can't see any black symbols on Fig 10
- Caption: downward / upward triangles: I can't see any of these on Fig. 10

Fig. SI 2:
- "The distributions consider cloudy…"

Fig. SI 3:
- caption L3: "observational data"

Fig. SI 4:
- I find the dark blue lines hard to distinguish in my printed copy

Fig. SI 7:
- 7b: where are the points for ctrl data on the CIN and CAPE Figures?

Fig SI 8:
- How sensitive is this figure to how you choose to define low / medium / deep cloud tops?
- It would be worth placing labels on the Figure with "low", "med", "deep" near the relevant set of points, just to make it clearer for the reader.
- Again, I don't think these points should be joined with lines.

Fig SI 9:
- I found it almost impossible to understand this Figure. Are condensation and deposition shown separately, or combined? What are the symbols? What do IG and IL refer to? Also, as mentioned in the major comments, I don't think that the points

representing the ensemble members should be joined with lines. This is not a continuous dataset. (My printed copy also has different linestyles in the Figure, which are not explained, but I suggest to remove the lines entirely).
- Caption: … and deposition rates

**Comments on tables:**

Table 1:
- Caption: variable (columns)… aerosol scenarios (rows) – aren't these the other way round? (Don't the rows show the variables and the columns the aerosol scenario?)
- What do the bold numbers in the table mean?

Table 2:
- Why does the low-high scenario need so few samples compared to low-standard or standard-high?

**Additional references:**

https://journals. Lee, S.S., 2012: Effect of Aerosol on Circulations and Precipitation in Deep Convective Clouds. *J. Atmos. Sci.,* **69**, 1957–1974, https://doi.org/10.1175/JAS-D-11-0111.1 .org/doi/abs/10.1175/JAS-D-11-0111.1

---

## Referee Comment (RC2) · Anonymous Referee #2 · 31 Mar 2018

The following is a review of the article entitled "Aerosol-cloud interactions in mixed-phase convective clouds. Part 2: Meteorological ensemble." by A. Miltenberger and coauthors for publication in *Atmospheric Chemistry and Physics* journal.

**General conclusion:**

I think the article is appropriate for *Atmospheric Chemistry and Physics*, because it is an interesting study which investigates a case of moderately deep convection developed along sea-breeze convergence over the southwestern peninsula of the UK, by the use of high-resolution model simulation ensembles, with the motivation of enlightening the relative contribution of variations in meteorological and aerosol initial and boundary conditions to different cloud properties.

From my point of view, the main article strength is that it is able to disentangle (to a certain extent) whether the cloud and precipitation effects are due to the meteorological variability or due to aerosol background concentration initial conditions, at least for the case study of mixed-phase convective clouds. Moreover, I think the choice of an increase and decrease in passive aerosol concentration by a factor of 10 is appropriate for the simulations with perturbed aerosol profiles. On the contrary, the main weakness is that there is too much description of the case study until the interesting conclusions are reached. That, at my understanding, makes the reading too much detailed and tedious to follow. Therefore, I would recommend to reduce the number of figures (or move them to the SI) and get to the point on the important findings and conclusions (basically sections 5, 6 and 7) sooner. Also, I would suggest to the authors to use the significance results presented in Table 1 (unpaired) all over the whole discussion text, since it has important implications whether a result is significative or not. For instance, in section *5.4.Radiation* I would add that only OSR results are significative (and only those regarding the comparison between low-standard and low-high aerosol concentrations) and not OLR results, thus the reader do not have to wait until the end of the paper (section 7) to know that some of those differences described before are in fact not significative. Besides, there are many other technical issues and questions which I am listing in the following *Specific comments* and *Typos*.

Ultimately, I recommend the article publication in ACP because it is clearly contributing to enhance the knowledge of the scientific community regarding to aerosol-cloud interactions in mixed-phase convective clouds, but a major revision is required because the paper could still be improved.

**Specific comments:**

Section 1: The introduction is appropriate since it is explaining the nowadays main issues, providing the necessary state-of-art, and introducing the contribution of the present study.

- Page 2, lines 6-9: however it is true that in the last decades was a large increase in anthropogenic aerosol emissions, I would add that "the emissions have decreased in the last decades (in comparison the 80s-90s maximums) thanks to the introduction of pollution policies in the developed countries in the Northern Hemisphere".

Section 2:

- Page 4, lines 20-27: the fact that 9 ensemble members were selected is repeated 3 times in only 8 lines, please consider rewriting the paragraph. Moreover, could you explain how they were chosen among the 33 global ensemble members? (see comment in Fig. 1)
- Page 5, line 11: please clarify the sentence "[…] nested simulations with a h a grid […]".
- Page 5, lines 13-14: consider the necessity of repeating the same set of references for CASIM since they have been cited already in page 3, lines 33-34.
- Page 5, lines 17-18: How and why moisture conservation is enforced?

Section 4:

- Page 10, lines 9-10: could you clarify why "the surface moisture flux adds some modifications to the boundary layer moisture budged, e.g. ensemble members 3 and 4, and members 7 and 8, respectively", because I am not able to see it in the figures.
- Page 10, line 11: from my interpretation of figure 4b, I think ensemble members cntl, 4 and 8 have a particularly large surface sensible heat flux.
- Page 10, lines 18-19: could you add references/citations to the cloud top height threshold based on the condensed water content, and to the cloud fraction based on the condensed water path?
- Page 10, line 22: Are not ensemble members 2 (instead of 8) and 5 (instead of 4) those with the largest and smallest mean cloud top height, respectively? (for standard aerosol case).
- Page 10, line 25: based on figure 4a, I would say that ensemble members 2, 4 and 8 are those with high surface moisture fluxes (and not 4 and 7). On the other hand only ensemble members 4 and 8 have larger low-cloud fraction.
- Page 10, line 28: based on Fig. 7d, I think that the variation in PE is higher than 5 %.
- Page 10, line 32: based on Fig. 7d, I think ensemble member 6 has a relatively large PE (instead of 8).
- Page 11, line 3: based on Fig. 11a, "the largest (smallest) values occur for ensemble member 1 (8)" (instead of 2 (7)).

- Page 11, lines 3-4: please, recheck the sentence regarding the relation between outgoing longwave radiation and the cloud top heights since for instance in Fig. 6 it is seen that ensemble members 4 and 5 have similar mean CHT but on the other hand large differences on OLR. How do you explain that?

Section 5:

- Page 11, line 5: why the section title says "identical meteorological initial and boundary conditions" when in fact here are discussed the differences between ensemble members with different meteorological initial and boundary conditions (as stated in line 13 in the same page)?
- Page 11, lines 24-25: does it mean that there are less clouds but larger?
- Page12, line 2: according to SI Fig. 8a, I think ensemble member number 4 is missing from the list of ensemble members where the change in low cloud top fraction is dominant.
- Page 12, lines 32-34: Is not ensemble member number 7 also fitting in the exception list? At least this is what I can see from Fig. 7c. For this reason, I recommend changing the graphic color palette or enlarging the figure. Anyway, what do you think is the reason why the control simulation has higher surface precipitation with the low aerosol scenario?
- Page 12 line 34 and page 13 line 1. Could you check the affirmation again? I do not see the comparatively large condensate gain in Figure 7a.
- Page 13, line 1: please check the following inconsistency: you stated "Accordingly, the precipitation increase for these members…" when you just said in page 12 line 34 that "[…] members 6 and 8 with no change in accumulated surface precipitation."
- Page 13, line 2: I do not see clear the sentence "For the other members, the change in PE dominates over changes in condensate production" because the change in PE is also large for ensemble members 6-8, and for some of the other members it is actually not that large.
- Page 13, line 3: consider adding "(decrease)" after "precipitation response".
- Page 13, line 10: I am not sure if ensemble member number 5 falls in this exception list, could you please check it again? Additionally, from figure 8a it is remarkable that for some ensemble members (3, 4 and 8) the mean precipitation turns to zero with the highest aerosol concentration scenario, perhaps you would like to highlight it into the discussion.
- Page 13, lines 12-14: I do not think "all percentiles up to and including the 75[th] percentile show an increase with the aerosol concentration" for all ensemble members, since in ensemble member 4 the standard aerosol is lower than the other two and in member 7 the high aerosol is lower than the low and standard aerosol concentration.
- Page 13, lines 16-17: consider adding to the discussion the fact that, as other studies have shown, enhanced aerosol scenarios suggest more freezing processes inside clouds and invigoration, provably due to longer cloud lifetime.
- Page 13, lines 33 and 34: I agree with the sentence "This change is consistent with the increased CDNC and small impact of the aerosol scenario on the cloud fraction", however,

the change (decrease) in the CF shown in Fig. 6 would cause the opposite effect. How do you explain that? (The same reasoning applies in the sentence in page 15, lines 7-8).

- Page 13, lines 30-34, and page 14, line 1-2: how do you know that radiative signal presented here is mainly due to CDNC changes and not due to an increase in aerosol scattering (the so-called 'direct effect')? Moreover, consider referring here to the 'indirect effect' or 'cloud albedo effect' and adding "Twomey, 1974" citation reference. Consider also adding "(increase)" after "due to CDNC changes".

Section 6:

- Page 14, line 16: I would rather prefer "changes follow a similar pattern for each meteorological ensemble member" than "changes are similar for each meteorological ensemble member".
- Page 14, lines 11-19: I miss the authors saying something regarding changes in WP and LWP in this paragraph.
- Page 14, lines 28-29: why the authors only consider the time frame from 9 to 19 UTC while the model was run from 0 to 24h on the 3/8/13? Is it due to meteorological reasons or because of the model spin-up time period? Moreover, why in Fig. 12 is used the time average 10 - 19UTC? Is it a typo?
- Page 14, line 29: please change "Figs. 5-11" for "Figs. 5-9 and 11". Moreover, is Fig. 10 done with the data from 0 to 24h or with data from 9 to 19h?
- Page 14, line 30: the variables are not plotted in a box-plot but in an error bar type plot.
- Page 14 line 35 and page 15, line 1: could it be related to a longer cloud lifetime?

Section 7:

- Page 16, lines 17-19: I suggest changing the sentence since some of the cloud properties stated here are poorly modified by the aerosol perturbations (e.g. cloud fraction), not modified considering all perturbations (e.g. condensation gain G is not significative for standard-high comparison), and not modified if unpaired cases are considered (e.g. precipitation rate or cloud fraction are only significative for paired cases)
- Page 16, lines 22-34: Could you explain in more detail how the significance analysis of paired and unpaired was done? How do you pair the ensemble members? I do not really see the advantage of the paired significances with so few ensemble members and it looks to me confusing, if not misleading, since as you already say in page 16, lines 26-27, the statistical analysis is based on a very small sample, which affects the validity of several assumptions. Therefore, I personally prefer the results with unpaired cases because the sample is already too small to be paired and because the results with unpaired cases better reflect the results and error bars (spread) shown in all figures and in particular in Fig 12, even at the expense of having less significative results.

Section 8:

- Page 18, lines 31-32: could you give an example of those "variables closely related to aerosol concentrations" and for those "variables that are linked to aerosol concentrations by a series of complex processes" which apply to the investigated case?

I would suggest changing the section 4, 5 and 6 titles since they are not helpful for understanding the article structure. In my opinion, it would be easier for the reader if they are rewritten somehow in that way:

- Section 4: Comparison of the cloud-properties results among 10 different meteorological ensemble members (unperturbed aerosol profiles)
- Section 5: Analysis of the results regarding aerosol-induced changes (3 aerosol concentration scenarios) among 10 ensemble members
- And section 6: Cloud-adjustments attribution (due to initial meteorological and boundary conditions or aerosol concentration loads)

References:

- Fan et al. 2016: has the DOI link repeated.
- Sheffield et al. 2015: has the DOI reference repeated?
- Tao et al. 2012: please check if the reference between the DOI and the year should be there.

Figures:

Generally speaking, the way the figures are presented is a bit chaotic. First of all, they are not correctly ordered, and secondly there are too many. I suggest the following improvements:

  o Re-organize all the figures in order of appearance in the text. For example: figures 6 and 7 are referenced in the text before 4 and 5 have been, and figure 11 before figure 10.
  o Remove linking lines between ensemble members from the following figures: 4b, 7a and b, SI 8a, and SI 9.
  o Consider changing the color palette for the different aerosol load runs in the following figures, since it is difficult to differentiate them: 4b, 7, SI 8a, and SI 9.

Comments on figures from the main discussion paper:

- o Fig. 1: in the caption it is stated that 9 ensemble members were chosen from 33. Could you give more information on that? How they were chosen? Which criteria were used?

   Could you change the x-axis ticks in a way that both graphs (a and b) have the same.

- o Fig. 3: please add the data information in the last column "mean" in c and d, otherwise remove it from the graphs.

- o Fig. 4: consider removing it or moving it to the SI since it is only cited once in the paper (and actually only Fig. 4a), the latent heat flux (Fig. 4b) is not used in for the discussion, and the sensible heat flux figure is only used once. Moreover, I think the caption is wrong because it does not match with any of the three graphs and legends.

- o Fig. 5: since it does not show big differences between ensemble members I would suggest moving it to the SI.

- o Fig. 7: as said before, I recommend changing the color palette of the graphic or enlarging the figure.

- o Fig. 10: please change the legend with the symbols that appear in the graph (i.e. there are no squares in the graph). Also the caption is wrong since there are no "black symbols", "downward pointing triangles" or "upward pointing triangles".

- o Fig. 12: this is a really interesting and helping figure. I just want to say that adding a legend or a caption explanation regarding the colors, as well as regarding the acronyms used in the x-axis, would improve it. Please, also include if CDNC is at the cloud base or at the top.

Comments on the SI figures:

- o SI Fig. 1: I would include this figure in the main paper (not in the supplement) since it helps the reader quickly identifying the region on the model simulations have been done and how they look like.

- o SI Fig. 4: it needs some improvements since it is not intuitive. I would suggest plotting in different colors the temperature and the dew temperature profiles, from both model and observational data.

- o SI Fig 6 and 7 captions: "The box plots represent the temporal variabilitiy of each variable" should read "The box plots represent the temporal variability of the variable" or "The box plots represent the temporal variability of the variable for each ensemble member".

- o SI Fig. 6: why ensemble 4 with low aerosol look so different from the others? Is there any apparent reason?

- o SI Fig. 7: CAPE for cntl simulation is missing. And please, either add values for the last column or remove "mean" from the graphic.

o SI Fig. 9: I suggest adding into the caption the description of IG and IL (from the legend) as well as "[…] condensation (C) and deposition rate (D) […]".

Tables:

- Table 1: as mentioned before, I would remove the unpaired results.

**Typos:**

- Page 4, line 17: "section ??" should read "section 5".
- Page 7, line 24 and line 28: "15 UTC" should read "15.20 UTC".
- Page 10, line 17: "coherent areas" or "consecutive areas"?
- Page 10, line 18: "arial fraction" should read "areal fraction".
- Page 11, line 9: the word "and" is missing between "lower" and "respectively".
- Page 11, line 13: the word "part" is missing between "first" and "of this study".
- Page 12, line 5: remove "is" from the sentence.
- Page 13, line 10: remove "also" from the sentence.
- Page 13, line 32: add the word "by" after "cloud top and".
- Page 13, line 34: change "increasing" by "increased".
- Page 14, line 11: "section" should read "sections".
- Page 18, line 21: "aerosol-induce" should read "aerosol-induced".

---

## Author Comment (AC1) · 5 Jun 2018

**Reply to comments from reviewer #1**

**General comments:**
*1. The manuscript contains many typos and errors that the team of authors should really have addressed together through proof-reading before submission for review. I have listed these in my specific comments.*

Reply : Thank you very much for pointing these out! We have corrected all the highlighted issues and proof-read the new manuscript more carefully.

*2. I believe there are some inconsistencies in the references to ensemble members. I have noted these in my comments on Section 4.2.*

Reply : We are very sorry that there were inconsistencies in ensemble member references in the original manuscript. We have very carefully checked they are ok in the new version.

*3. **Figure ordering:** I found the order of figures somewhat counterintuitive and hard to follow.*
*I had to lay my printed copy of the figures and the supplementary figures out next to each other in order to follow the arguments made in the text. Whilst I appreciate that there are*
*already a lot of figures in the paper, I would suggest moving figure S8 to the main paper if possible. Further, the figure ordering in the main paper is not logical. I appreciate that it is difficult to optimally order figures when investigating concurrent sensitivities, but I would recommend placing the order of current Figures 7,6,5 as such. On P10 L4, the reference to Fig 7a, I had to jump ahead several Figures in order to see this. On P10 L34 I had to jump back again to Fig. 5, which is referenced for the first time after Fig.s 7 and 6 are discussed in detail. Why put Fig 5 in its current location? You would make it much easier for the reader if it appeared after 6 and 7.*
*Page 11, 1st paragraph: again, you refer to Fig 9 and then immediately after to Fig 11, and Fig 10 is not even mentioned until page 13.*

Reply : Thank you for your suggestions. We have re-ordered the figures so that they appear in the sequence they are mentioned in the text. Also, we changed the partitioning of figures between the main paper and the SI to have only the most crucial figures in the main paper following your suggestions and those of reviewer 2. For example, former SI Fig 8a is moved to the main text (new Fig. 8 b).

*4. **General comments on figures:***
*Many of the figures have lines joining the points representing each ensemble member. This is misleading, as the abscissa on these figures show ensemble members (a discrete dataset) and not continuous data. I recommend removing these lines.*

Reply : We have removed these lines from all figures as suggested.

*Figure SI 9 – I tried very hard to understand this Figure, but it many things in it don't make sense to me. See notes under my comments referring to individual figures.*

Reply : We are were sorry for the poor description of this figure. The caption and labelling has been improved, so it should be comprehendible now.

*5. **Section 2:***
*Stochastic physics – are stochastic physics used in the regional model as well as the global model? Are stochastic physics used in the full set of ensemble runs? (Are you using stochastic physics as well as perturbed initial conditions?) What kind of stochastic physics are used? Which schemes and which parameters? Etc. This needs a little more explanation if you are discussing a study which aims to capture meteorological variability.*

Reply : Stochastic physics are only used for the re-run of the global operational ensemble and not in the regional ensemble simulations. This has been clarified in the model set-up description (section 2: p. 4, l. 23-25 & p. 5, l. 9-11). Since stochastic physics are only used to derive the perturbed initial and boundary conditions, they do not influence the actually used ensemble data

and we therefore refrain from a more detailed description of the stochastic physics for brevity (reference is provided for interested reader!).

**6. Section 3.1:**

*P6 L13 – Was the model microphysics output passed offline through the same radar algorithm as the Radarnet data? If not, could part of the difference be because the online UM dBZ calculation is different from the dBZ calculation in the Radarnet algorithm?*

Reply : No. The radar reflectivity is computed from the modelled hydrometeor properties online, i.e. within the model. For the modelled surface precipitation, we are using the direct model output, i.e. surface precipitation is not diagnosed from the modelled reflectivity fields. The model assumes only Rayleigh scattering and does assumes particles have a single phase, e.g. partially melted hydrometeors do not exist in the modelling world. Conversely, the radar algorithm does not account for sub-cloud evaporation of rain, which has been shown in previous work to affect retrieved surface precipitation rates. Therefore the differences, could be due not only to a deficiency in the model microphysics, but also to issues with the radar-derived surface precipitation or radar reflectivity calculation in the model.
We added the following to clarify this point (p. 6, l. 17-20):
"While the model derived surface precipitation is the sedimentation flux at the surface, the radar derived surface precipitation is computed from the low-level radar reflectivity according to Harrison et al. (2009). Accordingly, the modelled and radar-derived surface precipitation products involve different assumptions. Fore For example, sub-cloud evaporation is not taken into account in the retrieval of surface precipitation rates from observational data."

**7. Section 4.1:**

*P9 L5-6: "These members have a higher cloud fraction" – do you know why this is the case for these members?*

Reply : As discussed in section 4.2 these members have a higher boundary layer moisture convergence, which most likely explains the higher cloud fraction. The text has been modified as follows (p. 8, l. 27-30):
"Only in ensemble members 1 and 2 the temperature difference remains smaller than 1.5 K (SI Fig. 7 c). These members have a higher cloud fraction in the morning (not shown), which is likely related to a relatively large large-scale moisture convergence. The higher cloud fraction reduces radiative heating of the land surface explaining the smaller peak land-sea temperature difference."

*P9 L17-18: Is this also related to the cloudiness (higher cloud fraction in these members)?*

Reply : The smaller wind speed and more southerly wind direction should affect the propagation of the sea-breeze front inland and therefore changes the low-level convergence. However, this relation is not very strict, as e.g. ensemble member 5 to 8 also have similarly high low-level convergence. The latter do not have a particularly large cloud fraction. Therefore, we think the cloud fraction variability is dominated by the differences in large-scale convergence. Although of course differences in sea-breeze convergence strength impact cloud fraction, but these differences are much smaller than those in large-scale convergence.

**8. Section 4.2:**

*P10 L11-12: Members 4 and 7 have a particularly large surface sensible heat flux – can you explain why? It doesn't seem like they stand out in terms of cloudiness (Fig. 6).*

Reply : The high sensible heat flux for these members is a combination of high surface temperature and high surface wind speed (Fig. 1). The surface temperature in these ensemble members is already in the upper range at 9 UTC, i.e. before the onset of significant radiative heating. Also the relatively large surface wind-speed is consistent with the upstream profiles. Therefore the large sensible heat fluxes is likely related to changes in the initial and boundary conditions and not so much related to differences in cloudiness. Since the sensible heat flux is not discussed anymore in the revised version of the manuscript, there are no alterations regarding this issue in the text.

[Figure]

**Figure 1.** Mean surface temperature (top), 10 m wind velocity (middle) and sensible heat flux (bottom) for the different ensemble members.

*P10 L22: largest (smallest) values for ensemble members 8(4) – I find it hard to see by eye on this Figure, but doesn't this actually apply to members 9(5) not 8(4)?*

Reply : We are sorry for the incorrect referencing of ensemble members in the text. The reviewer is of course right and we have corrected the text (p. 9, l. 24; checking the numerical values indicates ensemble number 2 has the highest mean cloud top height very closely followed by ensemble member 9).

*P10 L22-23: Really? I find this hard to see (Fig 6d vs Fig SI 7c)*

Reply : The overall tendency in the mean equilibrium level pressure corresponds to the overall tendency in the mean cloud top height: For example, ensemble 5 with the smallest mean cloud top height has the largest equilibrium level pressure and ensemble 2/9 with the largest mean cloud top height have the smallest equilibrium level pressure. Modified text (p. 9, l. 23-26):

"Mean cloud top height varies by about 750 m between ensemble members (Fig.8) with largest (smallest) values for ensemble member \2 and 9 (5). Variations in mean cloud top height are in general consistent with those of the equilibrium level pressure (SI Fig. 10 c): For example, the equilibrium level pressure in ensemble member 5 is largest, while members 2 and 9 have the smallest equilibrium level pressure."

*P10 L25-26: ensemble members 1,2,5,8 have a relatively large fraction of deep clouds – I don't see this. What about e.g. member 6 (Fig SI 8a)?*

Reply : The reviewer is correct ensemble member 6 has the largest deep cloud fraction. This sentence is not part of the revised manuscript anymore, because we tried to shorten the manuscript as requested by reviewer 2.

*P10 L29: changes in condensate generation, i.e. air mass lifting – have you looked at the dynamical convergence to see if this is the case?*

Reply : We are not sure what the reviewer mean with "dynamical convergence". The large-scale boundary layer convergence and the low-level convergence have been discussed in section 4.1 and 4.2. Their variability corresponds in general very well with the variability in G.

*P10 L32:*
- *"member 8 has a relatively large PE" – I disagree with this. Many others have a greater PE, e.g. 1, ctrl, 6 (Fig. 7)*
- *"and the largest fraction of clouds with tops above 4.3 km" – I also disagree with this. The largest fraction of clouds with tops above 4.3 km is seen in member 6 (Fig SI 8a). I think in this sentence perhaps the authors mean to refer to member #6, not member #8? Then I agree with the statements made in the sentence.*

Reply : This statement refers indeed to ensemble member 6 and has been corrected accordingly (p. 9, l. 34).

*P11 L1-4: this final section is not particularly well-explained and no relevance is given. Can you say anything about the processes and impact or importance?*

Reply : We have expanded this section as follows (p. 10, l. 8-15):

"Mean reflected shortwave radiation ranges from 130 W m$^{-2}$ to 155 W m$^{-2}$ (Fig. 11 a). The reflected shortwave is influenced by the cloud cover and the cloud droplet number concentrations. The largest (smallest) outgoing shortwave flux is predicted for the ensemble members with the largest (smallest) cloud fraction, i.e. ensemble 1 (8). Since the CDNC variability is small (Fig. 5), the variations in cloud fraction between ensemble members is dominating the variability of outgoing shortwave radiation. Changes in outgoing longwave radiation are on the order of 3 W m$^{-2}$ (Fig. 11 b). The outgoing longwave radiation is influenced by the surface temperature, the cloud top height and the cloud fraction. While differences in the cloud top height distribution contribute to the variability in outgoing longwave radiation, variations in the clear sky outgoing longwave radiation dominate the overall variability due to the relatively small cloud fraction (SI Fig. 13)."

*P11 L3-4: Largest (smallest) values occur for ensemble members 2(7) – I only just agree with this. Do you mean member 1 not member 2 for the largest SW radiation values and largest CF?*

Reply : Yes, this is member 1. We changed the text accordingly (p. 10, l. 10).

***Specific comments and typos:***
*P1 L16: "consider" -> "considered"*
*P2 L3: "climate system"*
*P2 L4: "The main issues..."*
*P2 L5: "... on model grid scales several orders of magnitude larger, and the..." P2 L6: "In the last few decades" / "In recent decades"*
*P2 L6-7: "the modification of cloud properties has been studied in particular"*
*P2 L7-8: "... and the relation between particle number concentrations and radiation" – this whole sentence feels quite clumsy.*
*P2 L13: "necessitated by" -> "necessary because of"*
*P2 L15: "changes to simulated for individual clouds" – simulated what?*

Reply : Thank you for pointing these out. All issues have been fixed as suggested.

*P2 L17: You could also include a reference here to the 2012 paper by Seoung Soo Lee where placing an aerosol perturbation in the mesoscale domain of a simulation led to intensification of convection within an MCS but suppressed precipitation in the larger-scale domain. (Reference provided at the end of this set of comments)*

Reply : We have added this reference (p. 2, l. 14-19):
"These interactions can at least partly compensate the large changes simulated for individual clouds (e.g. Lee, 2012; Seifert et al. 2012). In a case-study of tropical deep convection, Lee (2012) found that locally invigorated convection in polluted conditions induces stronger large-scale subsidence resulting in an overall suppression of precipitation on a cloud-system scale. Seifert et al. (2012) demonstrated with simulations extending over three summer seasons that aerosol perturbations can produce large local changes in precipitation, while not significantly changing the mean precipitation."

*P2 L29: Southern Great Plains*
*P3 L6: What do the authors mean by "cloud-induced changes to large-scale forcing"? Does this refer to large-scale circulation and / or synoptic forcing, or something else?*
*P3 L8: "has also" -> "also has"*
*P3 L10: relay -> rely*
*P3 L11: rises -> raises*
*P3 L12: datasets*
*P3 L12: has recently been demonstrated*
*P3 L14: Southern Great Plains*
*P3 L23: in future forecasting systems*
*P3 L28: 30$^{th}$*
*P3 L34: baseline*

*P4 L2: a precipitation -> precipitation*
*P4 L4: the observed aerosol*

Reply : Thank you for pointing these out. All issues have been fixed as suggested.

*P4 L7: convective invigoration hypothesis needs a description and / or citation*

Reply : We added a reference to the Rosenfeld et al. (2008) paper (p. 4, l. 7)

*P4 L10: investigate whether the*
*P4 L17: Section number missing. (should this be Section 5?)*
*P4 L23-24: The way this sentence is written doesn't quite make sense.*
*P4 L25: do you mean "9 members are selected from", not "selected for"?*
*P4 L32: mesoscale*
*P5 L3-4: repetition of "current study"; you could just say "to our main conclusions".*
*P5 L6: In addition to (delete comma)*
*P5 L11: h a grid -> horizontal grid ?*
*P5 L32: datasets*
*P6 L3: peninsula (remove capital P)*
*P6 L16: have also reported*
*P6 L15: underdispersive over longer*
*P6 L31: smaller if (delete comma)*
*P7 L21, 23, 26: dewpoint*
*P7 L31: ensemble members*
*P9 L20: similar, with a well-mixed*
*P9 L25: temperate -> temperature*
*P10 L3,8: mesoscale*
*P10 L4: "G is very well correlated" – have you actually correlated this (or can you)?*
*P10 L4-5: Figures 4a and 7a are difficult to compare as they are on different pages*
*P10 L14: convergence*
*P10 L18: areal*
*P10 L26: an about 20% -> about a 20%*
*P10 L 10-31: refer to Fig SI 8a*
*P11 L3: largest (smallest) cloud fraction – please refer to Fig 6c.*
*P11 L4: distribution of cloud top heights (Fig. 11b) – you also need to refer to the Fig. showing CTH.*
*P 11 L8: "low", "high": open quotations are the wrong way wrong (LaTeX `` not "?)*
*P 11 L9: "which have a factor of 10 lower and higher aerosol number concentrations, respectively, than the standard profile"*
*P11 L10: altitudes*
*P11 L10: The mean and effective radius – mean what? Mean radius and effective radius? P11 L12: the first section of this study?*
*P11 L13: ensemble members*

Reply : Thank you for pointing these out. All issues have been fixed as suggested and sentences have been reformulated where necessary.

*P11 L17; Figure 5 should be moved, as discussed in the major comments*

Reply : The figures have been reordered (s. reply to general comment 3).

*P11 L21: "suggest only minor changes in the cloud-base vertical velocity distribution" – can you plot this distribution? Doesn't this contradict the previous statement made about convergence?*

Reply : The cloud-base vertical velocity distribution is shown in Fig. 2 (new SI Fig. 9b). This confirms our hypothesis of small changes in the cloud-base vertical velocity distribution. While these changes are small, the variability in average cloud-base updraft between ensemble members is still larger than the difference in average boundary-layer top moisture content, which is

[Figure]

**Figure 2.** Cloud base vertical velocity distribution for the simulations with the standard aerosol profile considering all grid points at cloud base and a positive vertical velocity for the time period 09 - 19 UTC.

all that we claimed earlier. Note that the discussion of the latter has been removed from the manuscript to meet demands for shortening the text from reviewer 2.

*P11 L24: "the number of cells decreases with increasing background aerosol concentration, but the cell area increases" – this is interesting! Can you explain why this happens?*

Reply : We can only speculate about the physical reason for this behaviour, which has been done in the first part of the study. We added a brief summary of the hypothesis the revised version (p. 11, l. 8-11): "It has been hypothesised in the first part of this study, that the slower conversion of condensate to precipitation in high aerosol conditions allows clouds to grow larger and merging with other updraft cores resulting in overall fewer, but larger clouds. Also, energetic constraints potentially limit an increase in overall lifting and cloud fraction. "

*P11 L30: cloud top height increases*
*P12 L2: "ensemble members 1,2,7,8, and 9" (missing space between "and9")*

Reply : Thank you for pointing these out. All issues have been fixed as suggested.

*P12 L2: "ensemble members 1,2,7,8, and 9" – this is also true for member 4*

Reply : Sorry again for the confusion with the ensemble member numbers. This list should read 1, 2, 4, 7 and 9 and has been corrected accordingly (p. 11, l. 18-19).

*P12 L3: "does not increase further (members 1 and 2)" – doesn't member 2 increase?*

Reply : The median mean cloud top height does, while the time-average mean cloud top height does not increase from the standard to the high aerosol run. We have clarified this in the text (p. 11, l. 18-19).

*P12 L5: higher than 4.3 km shows only*
*P12 L7: aerosol scenario is likely (remove comma)*
*P12 L8: "maximum" (open quotation incorrect way round)*
*P12 L12: "only a small change"*
*P12 L14: "-4 – 2.5%" - this notation is confusing. Do you mean -4% to -2.5%, or -4% to +2.5%?*

Reply : Thank you for pointing these out. All issues have been fixed as suggested.

*P12 L18: I do not understand Figure SI9.*

Reply : We apologise for the poor presentation of this figure. We have improved the legend and axis labels in the figure as well as the caption (new SI Fig. 14 d).

*P12 L21: Can you plot delta G and delta L instead of G, L?*

Reply : We could plot ΔG and ΔL instead of G and L, which would make it easier to discern aerosol-induced changes. However, this would make the plot less useful to understand the meteorological variability. ΔG and ΔL were/are shown in previous Fig. 10 (new Fig. 13). We add a plot showing ΔG and ΔL in the format of previous Fig. 7b in the SI (SI Fig. 14 b and 15 b).

P12 L29: "seized" -> "sized"
P 12 L32: simulations in the standard
Reply : Thank you for pointing these out. All issues have been fixed as suggested.

*P12 L34: in Figure 7c in my printed copy, member 7 also looks like it has no change*

Reply : The change in precipitation for ensemble member 7 is very small. The numeric values indicate that accumulated precipitation slightly decreases when aerosol concentrations are increased from the low to the standard scenario. This is consistent with the position of ensemble member 7 off the one-to-one line in Fig. 13. In contrast, ensemble members 6 and 8 fall almost exactly on the one-to-one line in Fig. 13.

*P12 L34: "The latter have a relatively small decrease of PE and comparatively large delta G" – but member 1 also has a decrease in PE and delta G, but a decrease in precip in standard vs low scenarios, and is outside the shading in Figure 10.*

Reply : The four ensemble members with the smallest change in PE (from low to standard) are the control, member 6, member 1 and member 8, in this sequence. ΔG for ensemble member 1 is the the second smallest (joint position with ensemble member 3). In contrast, ensemble member 8, which has a similar change in PE, has the third largest ΔG. Also note that changes in PE operate on G, which is much larger in ensemble member 1 than 8, and not on ΔG. Hence, a relatively small change in PE is significant for ensemble member 1, while ΔG still dominates for ensemble member 8 despite a similar absolute change in PE. The text has been modified to make this clearer (p. 12, l. 18-21):
"Exceptions are the control simulation with a small increase in precipitation and ensemble members 6 and 8 with no change in accumulated surface precipitation. These ensemble members have a relatively small decrease of PE as well as a relatively large ΔG and G compared to ensemble members with a similar change in PE (e.g. compare ensemble member 1 and 8)."

*P13 L1: "comparatively large delta G" – this is hard to see from Fig 7a. Can you plot delta G and delta L instead of G and L?*

Reply : ΔG and ΔL are shown in previous Fig. 10 (new Fig. 13). We also added a figure showing ΔG and ΔL in the format of previous Fig. 7 to the SI. See also reply to comment on P12 L21.

*P13 L10: "Exceptions are ensemble members 3,4 and 5..." – you should point out that the behavior in each of these members is different from each other. For the (a) low to standard and (b) standard to high aerosol scenarios, member 3 has an (a) decrease and (b) increase, member 4 has an (a) decrease and (b) decrease, and member 5 has an (a) increase and (b) decrease.*

Reply : The text has been modified to: "Only in ensemble member 3 does the mean precipitation rate not decrease further in the high aerosol scenario, while in ensemble members 4 and 5 the decrease between the standard and the high aerosol scenario is comparable to the decrease between the low and standard scenario." (p. 12, l. 30-32)

*P14 L11: two sections*
*P14 L20: datasets*
*P14 L29: realisations*
*P15 L4: distribution in different cloud top height classes*
*P 15 L11: Precipitation formation is known...*
*P1 L21: accordingly displays*
*P15 L29: very similarly to*

Reply : Thank you for pointing these out. All issues have been fixed as suggested.

*P15 L34-35: "the liquid water path (...) shows little sensitivity to the aerosol scenario" – actually, there is a decrease in LWP (Fig 9b) which is not that much weaker than the increase in CWP (Fig 9a) – this indicates even more strongly than you currently state that the FWP must increase!*

Reply : The text has been modified to read: "However, the liquid waterpath (condensate in the cloud and rain category) shows relatively little sensitivity in its median value, while the mean liquid water path generally decreases with increasing aerosol concentrations." (p. 15, l. 29-31)

*P16 L9: mesoscale*
*P17 L2: "perfect" (open quotations incorrect way round)*
*P17 L3: only slightly different*
*P18 L4: exact number is*
*P18 L7: several 100 ensemble members -> several hundreds of ensemble members*

Reply : Thank you for pointing these out. All issues have been fixed as suggested.

*P18 L11: Why is low-high so different from low-standard and standard-high?*

Reply : This is mainly because the aerosol perturbation is a factor 10 larger, if the low and high aerosol scenario are considered, than in either low to standard or standard to high scenario. The larger amplitude in the aerosol perturbations results in larger aerosol-induced changes. However, the meteorological variability is in all combinations the same. Accordingly, the "signal-to-noise" ratio is larger in the low-high combination than any other. This has been included in the discussion: "The number of samples required depends on the amplitude of the aerosol perturbation (low and high aerosol scenario versus low/high and standard scenario) as well as the location in the aerosol space (different for increase or decrease relative to the standard aerosol scenario)." (p. 18, l.8-10)

*P18 L13: Accumulated precip stands out here – are you able to explaim why? (It's the only one that needs fewer observations for an increase of number concentration above the standard scenario).*

Reply : For all considered variables except P the aerosol-induced changes are smaller (or identical for CDNC) for an increase of aerosol concentrations above the standard aerosol scenario than for decreasing aerosol concentrations. In part 1, it is hypothesised that thermodynamical constraints lead to the saturation of the aerosol effect for high aerosol conditions. In contrast, for accumulated precipitation the aerosol-induced change increases with increasing aerosol concentrations. This is primarily due to PE changes. In part 1, we hypothesis that this change in PE is due to a larger export of condensate into the stratiform region with less active microphysics prompted by the thermodynamic limitations on cloud top height.
The text has been modified to (p. 18, l. 10-15): "In general, more observations are required for an increase of aerosol number concentrations above the standard scenario, which is related to the thermodynamic constraints on aerosol-induced changes in the considered case discussed in Miltenberger et al. (2018). The only exception is accumulated surface precipitation, for which fewer observations are required for an increase above the standard scenario. This reflects the larger aerosol-induced signal in accumulated precipitation for increased compared to decreased aerosol concentrations."

*P18 L14: "the thermodynamic constraints on aerosol-induces changes..." – constraints for this particular case, or general constraints?*

Reply : While there are very likely thermodynamic constraints on aerosol-induced changes in many situations, the conclusions are of course only valid for the investigated case. We have modified the sentence to reflect this (p. 18, l. 12).

*P18 L15: allows us to put the aerosol-induced changes / allows the aerosol-induced changes to be put*

Reply : Thank you for pointing this out. Fixed as suggested.

**Comments on individual figures:**
*Fig 4: Don't join these points with lines*

Reply : Done.

*Fig. 5:*
- *Can you comment on why these are so invariant?*

Reply : As discussed on p. 9, l. 15-17 of the manuscript the CDNC is not very variable across the ensemble, because the aerosol concentrations are identical in all members and the cloud base vertical velocity distribution in the different ensemble members is not strongly differing (see also Fig. 2 in this reply).

- *What do the colours represent?*

Reply : Added legend.

- *I think Figure 5 should appear AFTER Figure 6 (given the ordering of discussion in the manuscript)*

Reply : see reply to general comment 3.

*Fig. 6: "cloud fraction is the fraction of the domain for which" (add "the", remove comma)*

Reply : Done.

*Fig 7:*
- *Fig. 7a would be clearer if you plotted delta G and delta L instead of absolute values*

Reply : We could plot $\Delta G$ and $\Delta L$ instead of G and L, which would make it easier to discern aerosol-induced changes. However, this would make the plot less useful to understand the meteorological variability. $\Delta G$ and $\Delta L$ were/are shown in previous Fig. 10 (new Fig. 13). We add a plot showing $\Delta G$ and $\Delta L$ in the format of previous Fig. 7b in the SI (SI Fig. 14b and 15b).

- *Don't join these points with lines*
- *"the last column in each panel"*
- *Caption: "means" – do you mean ensemble mean, or ensemble means?*

Reply : Done. The caption has been modified to clarify the raised point.

*Fig 10:*
- *What do the open versus the filled circles represent?*
- *Legend: should "high processing" be "high aerosol"?*
- *Caption: "black symbols" – I can't see any black symbols on Fig 10*
- *Caption: downward / upward triangles: I can't see any of these on Fig. 10*

Reply : Done. The caption has been modified to clarify the raised points.

*Fig. SI 2: "The distributions consider cloudy..."*
*Fig. SI 3: caption L3: "observational data"*

Reply : Done.

*Fig. SI 4: I find the dark blue lines hard to distinguish in my printed copy*

Reply : The colour scheme has been adapted.

*Fig. SI 7: 7b: where are the points for ctrl data on the CIN and CAPE Figures?*

Reply : Thanks for spotting this. The data was actually missing from the plot. It has been included in the revised version.

[Figure]

**Figure 3.** Cloud top height distribution (a) and fractions of cloud top in specific altitude ranges (b-d). The thresholds for these ranges are modified by ±500 m in (c) and (d) compared to those used in the manuscript and (b).

*Fig SI 8:*

- *How sensitive is this figure to how you choose to define low / medium / deep cloud tops?*

Reply : The altitude bands were chosen to reflect cloud top ranges with a different response to aerosol perturbations (Fig. 3 a). The overall behaviour of the cloud top height fractions does not differ significantly for variations by ± 500 m in the thresholds (Fig. 3 b-d), although the absolute fraction values of course do.

- *It would be worth placing labels on the Figure with "low", "med", "deep" near the relevant set of points, just to make it clearer for the reader.*
- *Again, I don't think these points should be joined with lines.*

Reply : Done.

*Fig SI 9:*

- *I found it almost impossible to understand this Figure. Are condensation and deposition shown separately, or combined? What are the symbols? What do IG and IL refer to? Also, as mentioned in the major comments, I don't think that the points representing the ensemble members should be joined with lines. This is not a continuous dataset. (My printed copy also has different linestyles in the Figure, which are not explained, but I suggest to remove the lines entirely).*

Reply : We have improved the legend, axis labels and caption.

- *Caption: ... and deposition rates*

**Comments on tables:**

*Table 1:*

- *Caption: variable (columns)... aerosol scenarios (rows) – aren't these the other way round? (Don't the rows show the variables and the columns the aerosol scenario?)*
- *What do the bold numbers in the table mean?*

Reply : The caption has been modified accordingly.

*Table 2:*

*Why does the low-high scenario need so few samples compared to low-standard or standard-high?*

Reply : see reply to specific comment on P18 L11

---

## Author Comment (AC2) · 5 Jun 2018

**Reply to comments from reviewer #2**

*From my point of view, the main article strength is that it is able to disentangle (to a certain extent) whether the cloud and precipitation effects are due to the meteorological variability or due to aerosol background concentration initial conditions, at least for the case study of mixed- phase convective clouds. Moreover, I think the choice of an increase and decrease in passive aerosol concentration by a factor of 10 is appropriate for the simulations with perturbed aerosol profiles. On the contrary, the main weakness is that there is too much description of the case study until the interesting conclusions are reached. That, at my understanding, makes the reading too much detailed and tedious to follow. Therefore, I would recommend to reduce the number of figures (or move them to the SI) and get to the point on the important findings and conclusions (basically sections 5, 6 and 7) sooner.*

Reply : The figures have been reconsidered and there have been significant changes to the distribution of figures between the main text and the SI as well as to the ordering of the figures. All figures are now ordered according to their mentioning in the text. Sections 3 and 4 have been shortened to streamline the text and focus on the key results. However, these sections are still important for the understanding of the results and for providing context on the ensemble performance as well as the magnitude of the meteorological changes, so we believe these sections belong into the manuscript

*Also, I would suggest to the authors to use the significance results presented in Table 1 (unpaired) all over the whole discussion text, since it has important implications whether a result is significative or not. For instance, in section 5.4. Radiation I would add that only OSR results are significative (and only those regarding the comparison between low- standard and low-high aerosol concentrations) and not OLR results, thus the reader do not have to wait until the end of the paper (section 7) to know that some of those differences described before are in fact not significative.*

Reply : We have included the statistical analysis in the text in section 5 and 6. These sections contain now references to Table 1.

*Besides, there are many other technical issues and questions which I am listing in the following Specific comments and Typos.*

Reply : Thank you very much for pointing these out! We have corrected all the highlighted issues and proof-read the new manuscript more carefully.

***Specific comments:***
*Section 1: The introduction is appropriate since it is explaining the nowadays main issues, providing the necessary state-of-art, and introducing the contribution of the present study.*
- *Page 2, lines 6-9: however it is true that in the last decades was a large increase in anthropogenic aerosol emissions, I would add that "the emissions have decreased in the last decades (in comparison the 80s-90s maximums) thanks to the introduction of pollution policies in the developed countries in the Northern Hemisphere".*

Reply : Thank you for pointing this out. For sake of brevity, we modified the text to say that anthropogenic aerosol emissions have changed significantly over historic period without specifying any trends (p. 2, l. 6-8): "In recent decades, the modification of cloud properties by aerosols has received particular attention, as anthropogenic aerosol emissions have changed strongly over the historic period."

*Section 2:*
- *Page 4, lines 20-27: the fact that 9 ensemble members were selected is repeated 3 times in only 8 lines, please consider rewriting the paragraph. Moreover, could you explain how they were chosen among the 33 global ensemble members? (see comment in Fig. 1)*

Reply : The selection procedure was already described on p. 4, l. 20-27. We have reformulated this paragraph to make the description clearer and to make the text more concise (p. 4, l. 27-31):

"The selection of the ensemble members for dynamical downscaling is based on the time-series of moisture convergence and moist static energy convergence computed over the regional model domain from the global model fields (Fig. 1). These timeseries are then used to construct a similarity matrix by summing the Euclidean distances of moisture convergence and moist static energy convergence. Using the the algorithm by J. H. Ward (1963) 9 clusters are defined and from each cluster the member closest to the mean cluster time series is chosen for downscaling."

- *Page 5, line 11: please clarify the sentence "[...] nested simulations with a h a grid [...]".*

Reply : Sentence was reformulated.

- *Page 5, lines 13-14: consider the necessity of repeating the same set of references for CASIM since they have been cited already in page 3, lines 33-34.*

Reply : We have removed these references here.

- *Page 5, lines 17-18: How and why moisture conservation is enforced?*
Reply : The methodology for moisture conservation is described in detail in the two cited papers by Aranami et al.. It is beyond the scope of this paper to provide a description of this methodology.

*Section 4:*
- *Page 10, lines 9-10: could you clarify why "the surface moisture flux adds some modifications to the boundary layer moisture budged, e.g. ensemble members 3 and 4, and members 7 and 8, respectively", because I am not able to see it in the figures.*

Reply : If the surface moisture flux is included the order of ensemble members (if ordering from largest to smallest moisture flux) changes. For example, the lateral PBL moisture flux (red points in Fig. 4 of the manuscript) suggests ensemble member 3 has a larger moisture flux than member 4. However, the surface moisture flux in member 4 is larger than that of member 3 (green symbols in the same figure). Hence, if the total boundary layer moisture flux is considered member 4 has a larger flux then member 3. The text has been modified to (p. 8, l. 15-16):
"The surface moisture flux adds some modifications to the boundary layer moisture budget, e.g. compare total and and lateral moisture convergence for ensemble members 3 and 4 and member 7 and 8, respectively."

- *Page 10, line 11: from my interpretation of figure 4b, I think ensemble members cntl, 4 and 8 have a particularly large surface sensible heat flux.*

Reply : We are really sorry that in several instance throughout the paper the wrong ensemble member numbers were referred to in the text. This is one of the instances. The reviewer is of course right that here it should read control, ensemble member 4 and 8 (instead of members 4 and 7). The section on surface heat fluxes has been removed from the manuscript to shorten the paper as suggested by the reviewer, so these changes are not actually applied in the new manuscript.

- *Page 10, lines 18-19: could you add references/citations to the cloud top height threshold based on the condensed water content, and to the cloud fraction based on the condensed water path?*

Reply : The chosen threshold for cloud top height (condensed water content larger than $10^{-6}$ kg kg$^{-1}$) is typically used in modelling studies to reflect detectability in observational studies and avoid issues with very small numeric values in models (e.g. Fridlind et al. 2010). The condensed waterpath threshold ($10^{-3}$ kg m$^{-2}$) is derived from this value: For a column to be classified as cloudy the minimum condensed water content needs over about 1000 m altitude range, i.e. for about 10 model levels (all considerations based on low-level values of gridspacing and atmospheric density). The condensed water path threshold is below the estimated lower detection limit of microwave satellite instruments (0.02 kg m-2, Grosvenor et al. 2017). These references have been added to the revised version of the manuscript (p. 9, l. 20-21).

- *Page 10, line 22: Are not ensemble members 2 (instead of 8) and 5 (instead of 4) those with the largest and smallest mean cloud top height, respectively? (for standard aerosol case).*

Reply : Yes, of course. The text has been corrected accordingly (p. 9, l. 24).

- *Page 10, line 25: based on figure 4a, I would say that ensemble members 2, 4 and 8 are those with high surface moisture fluxes (and not 4 and 7). On the other hand only ensemble members 4 and 8 have larger low-cloud fraction.*

Reply : The reviewer is right, this should refer to ensemble members 4 and 8. This section is not part of the revised manuscript anymore.

- *Page 10, line 28: based on Fig. 7d, I think that the variation in PE is higher than 5 %.*

Reply : We mean the difference between any to PE values does not exceed 0.05. We recognise that the percent notation introduces confusion. To avoid this confusion and address comments from reviewer 1 regarding this sentence, the new text reads: "In contrast, PE does not vary systematically with the large-scale convergence (Fig. 9 b)." (p. 9, l. 30-31)

[Figure]

**Figure 1.** Mean outgoing longwave radiation for cloudy grid points (top left) as well as mean cloud top height (bottom left) and mean cloud fraction (bottom right). The panels on the right show the total mean outgoing longwave (top) and outgoing longwave from cloudy (middle) and clear sky (bottom) gridpoints. Only simulations with the standard aerosol conditions are shown.

- *Page 10, line 32: based on Fig. 7d, I think ensemble member 6 has a relatively large PE (instead of 8).*

Reply : Yes, this has been corrected (p. 9, l. 34).

- *Page 11, line 3: based on Fig. 11a, "the largest (smallest) values occur for ensemble member 1 (8)" (instead of 2 (7)).*

Reply : Yes, this has been corrected (p. 10, l. 10).

*Section 5:*
- *Page 11, lines 3-4: please, recheck the sentence regarding the relation between outgoing longwave radiation and the cloud top heights since for instance in Fig. 6 it is seen that ensemble members 4 and 5 have similar mean CHT but on the other hand large differences on OLR. How do you explain that?*

Reply : Thank you for pointing this out. A more careful analysis shows that there is a relatively good correspondence of the mean OLR from cloud grid points and the mean cloud top height (left panels of Fig. 1). The match is of course not perfect, as OLR is very sensitive to cloud top temperature ($\sim T^4$) and therefore changes in the distribution matter as well, which are not reflected in considering the mean cloud top height. There is also a relatively strong variation in the clear sky

outgoing long wave radiation, which is caused by different surface temperatures, cloud positions, e.g. more or less cloud over the ocean, and water vapour path in the different ensemble members (Fig. 1, right panels). The domain average outgoing longwave radiation is a combination of these two contributions, but is strongly weighted towards the clear sky outgoing longwave due to the overall small cloud fraction ($\lesssim$ 0.15, Fig. 2).

We have included Fig. 2 in the SI and altered the text as follows (p. 10, l. 13-15): "While differences in the cloud top height distribution contribute to the variability in outgoing longwave radiation, variations in the clear sky outgoing longwave radiation dominate the overall variability due to the relatively small cloud fraction (SI Fig. 13)."

- *Page 11, line 5: why the section title says "identical meteorological initial and boundary conditions" when in fact here are discussed the differences between ensemble members with different meteorological initial and boundary conditions (as stated in line 13 in the same page)?*

Reply : We acknowledge that the section title is misleading. This section discusses aerosol-induced changes in the high and low aerosol scenario relative to the ensemble member with the same meteorological and the standard aerosol scenario. We have renamed the section "Aerosol-induced cloud property changes in different meteorological ensemble members (paired meteorology)"

- *Page 11, lines 24-25: does it mean that there are less clouds but larger?*

Reply : Yes exactly. This is included in the revised text (including also changes made in response to reviewer 1): "It has been hypothesised in the first part of this study, that the slower conversion of condensate to precipitation in high aerosol conditions allows clouds to grow larger and merging with other updraft cores resulting in **overall fewer, but larger clouds**." (p. 11, l. 7-9)

- *Page12, line 2: according to SI Fig. 8a, I think ensemble member number 4 is missing from the list of ensemble members where the change in low cloud top fraction is dominant.*

Reply : Yes, this has been corrected. (p. 11, l. 15-17): "In the control run and ensemble members 4 and 6, the decrease of the mean cloud top height is due to a reduction in the medium altitude fraction, while in all other members changes in the low cloud top fraction dominate."

- *Page 12, lines 32-34: Is not ensemble member number 7 also fitting in the exception list? At least this is what I can see from Fig. 7c. For this reason, I recommend changing the graphic color palette or enlarging the figure. Anyway, what do you think is the reason why the control simulation has higher surface precipitation with the low aerosol scenario?*

Reply : In ensemble member 7 surface precipitation is decreasing slightly. We agree this is hardly visible from the previous Fig. 7c, but it is clear from its position of the one-to-one line in previous Fig. 10. Previous Fig. 7c has been moved to the appendix and includes now a plot of $\Delta$P, which should make this even clearer (new SI Fig. 11).

As for the increase in the surface precipitation in the control simulation: We think this is due to the relative large increase in G (largest of all ensemble members) and the increase in PE. In fact, the control is the only ensemble member, for which PE increases from the low to the standard aerosol scenario. With out a detailed analysis of the thermodynamic, latent heating, and hydrometeor profiles similar to Miltenberger et al. (2018), it is difficult to speculate on the physical processes driving these changes. However, such an analysis is beyond the scope of the paper.

- *Page 12 line 34 and page 13 line 1. Could you check the affirmation again? I do not see the comparatively large condensate gain in Figure 7a.*

Reply : This point has not been well made in the previous manuscript, the text has been altered to: "These ensemble members have a relatively small decrease of PE as well as a relatively large $\Delta$G andG compared to ensemble members with a similar change in PE (e.g. compare ensemble member 1 and 8)." (p. 12, l. 19-21)

[Figure]

**Figure 2.** Difference in accumulated precipitation between the low and high aerosol scenario, respectively, and standard scenario.

*-Page 13, line 1: please check the following inconsistency: you stated "Accordingly, the precipitation increase for these members..." when you just said in page 12 line 34 that "[...] members 6 and 8 with no change in accumulated surface precipitation."*

Reply : If the numeric values are considered all have a slight increase in precipitation (Fig. 2). This figure has been included in the SI. However, changes in member 6 and 8 are very small, so we have changed the text to: "Accordingly, the precipitation response in these cases is either dominated by ΔG (control) or ΔG and ΔPE are of equal importance (member 6 and 8), as also indicated by their position in the shaded area in Fig. 12." (p. 12, l. 21-23)

*- Page 13, line 2: I do not see clear the sentence "For the other members, the change in PE dominates over changes in condensate production" because the change in PE is also large for ensemble members 6-8, and for some of the other members it is actually not that large.*

Reply : The amplitude of the ΔPE and ΔG does not in alone indicate, which one is more important, since ΔPE operates on G and not on ΔG only. The part of the ΔG - ΔL parameter space, in which changes in ΔG dominate is highlighted by the shaded areas in new Fig. 13. The derivation for this is presented in Appendix A of Miltenberger et al. (2018). All points except those representing the control and ensemble members 6 and 8 fall clearly outside this area.
We have clarified the basis for our conclusion in the revised manuscript: "This response is in all ensemble members dominated by PE changes (points outside the shaded area)." (p. 12, l. 25/26)

*- Page 13, line 3: consider adding "(decrease)" after "precipitation response".*

Reply : The sentence has been altered to "If the aerosol concentration is enhanced beyond the standard scenario, the precipitation decreases in all ensemble members (points above the one-to-one line)." (p. 12, l. 24/25)

*- Page 13, line 10: I am not sure if ensemble member number 5 falls in this exception list, could you please check it again? Additionally, from figure 8a it is remarkable that for some ensemble members (3, 4 and 8) the mean precipitation turns to zero with the highest aerosol concentration scenario, perhaps you would like to highlight it into the discussion.*

Reply : We think ensemble member 5 belongs in the exception list, as the precipitation change is almost symmetric for an increase or decrease of aerosol concentrations (Fig. 4). The text has been modified to make this clearer: "Only in ensemble member 3 does the mean precipitation rate not decrease further in the high aerosol scenario, while in ensemble members 4 and 5 the decrease between the standard and the high aerosol scenario is comparable to the decrease between the low and standard scenario." (p. 12, l. 30-31)
Thanks for pointing out that 75th percentile of mean accumulated precipitation is turning zero on some of the ensemble members for the high aerosol scenario. Since the distribution shown in these plots represents the temporal variability, this indicates an increased frequency of 10 min intervals without significant precipitation. The

[Figure]

**Figure 3.** Change in the average mean precipitation rate (cyan: low - std aerosol scenario, green: std-high aerosol scenario).

[Figure]

**Figure 4.** Time series of mean precipitation for ensemble member 1 (top left), 3 (top right), 4 (bottom left) and 8 (bottom right) for all three aerosol scenarios (cyan: low, blue: standard, green: high).

complete suppression of precipitation in members 3, 4 and 8 is due to much later onset of surface precipitation (Fig. 4). Note that higher percentiles in most runs are zero in most of the ensemble membersAs discussed for the control run in the first part of the presented study, clouds are mostly have lower cloud tops during the morning period and aerosol-induced changes have been found to be larger in this period. As this is extensively discussed in Miltenberger et al. (2018), we do not include this here. In particular, as splitting the response in different time periods would make the article more lengthy, while reviewers have already asked to shorten the manuscript.

- *Page 13, lines 12-14: I do not think "all percentiles up to and including the 75th percentile show an increase with the aerosol concentration" for all ensemble members, since in ensemble member 4 the standard aerosol is lower than the other two and in member 7 the high aerosol is lower than the low and standard aerosol concentration.*

Reply : The reviewer is right. The text has been modified accordingly (p. 12. l. 33-34): "All percentiles up to and including the 75th percentile show an increase from the low to the high aerosol concentration, while the 99th percentiles are generally smallest (largest) for the high (standard) aerosol scenario."

- *Page 13, lines 16-17: consider adding to the discussion the fact that, as other studies have shown, enhanced aerosol scenarios suggest more freezing processes inside clouds and invigoration, provably due to longer cloud lifetime.*

Reply : In the first part of the study, we have shown that at least for simulations very similar to the control run enhanced freezing does not contribute to convective invigoration in the investigated clouds. We therefore refrain from citing the convective invigoration hypothesis based on enhanced freezing here.

- *Page 13, lines 33 and 34: I agree with the sentence "This change is consistent with the increased CDNC and small impact of the aerosol scenario on the cloud fraction", however, the change (decrease) in the CF shown in Fig. 6 would cause the opposite effect. How do you explain that? (The same reasoning applies in the sentence in page 15, lines 7-8).*

Reply : The changes in cloud fraction (between 2 and 9 % depending on ensemble member) are very small compared to the changes in CDNC (factor ~7). Therefore the latter dominate the change in reflected shortwave radiation. The revised manuscript explains this in more detail (p. 13, l. 20-23):
"This change is consistent with the aerosol-induced change in CDNC and the cloud albedo effect (Twomey, 1977). The co-occurring decrease of cloud fraction under high aerosol conditions (between 2 and 9 % for a factor 10 aerosol change) counteracts the CDNC effect, but the cloud albedo effect dominates due to the large amplitude of the CDNC change (about a factor 7 for a factor 10 aerosol change)."

- *Page 13, lines 30-34, and page 14, line 1-2: how do you know that radiative signal presented here is mainly due to CDNC changes and not due to an increase in aerosol scattering (the so-called 'direct effect')? Moreover, consider referring here to the 'indirect effect' or 'cloud albedo effect' and adding "Twomey, 1974" citation reference. Consider also adding "(increase)" after "due to CDNC changes".*

Reply : The aerosol direct effect is not included in the model simulations. We have added the suggested reference and altered the text according to the other suggests (see reply to previous comment, p. 13, l. 20-23).

*Section 6:*
- *Page 14, line 16: I would rather prefer "changes follow a similar pattern for each meteorological ensemble member" than "changes are similar for each meteorological ensemble member".*

Reply : Thank you for this suggestion. The text has been altered accordingly (p. 14, l. 7).

- *Page 14, lines 11-19: I miss the authors saying something regarding changes in WP and LWP in this paragraph.*

Reply : Change in WP and LWP are discussed on p. 15, l. 33ff (old manuscript) and  p. 15, l. 28-33 (revised manuscript).

- *Page 14, lines 28-29: why the authors only consider the time frame from 9 to 19 UTC while the model was run from 0 to 24h on the 3/8/13? Is it due to meteorological reasons or because of the model spin-up time period? Moreover, why in Fig. 12 is used the time average 10 - 19UTC? Is it a typo?*

Reply : We only use the model output between 9 and 19 UTC, since this is the time period of main convective activity. Also, the first few hours of the simulation may be affected by model spin-up and later on some high-level cirrus clouds are advected into the domain, which influence the domain integrated cloud variables. We do not want to incorporate these, as they are not related to the convective clouds along the sea-breeze convergence and are mainly dominated by the boundary conditions for the innermost nest.
Fig. 12 actually shows the cloud properties for the 9 to 19 UTC time frame as all other plots. The caption has been corrected accordingly.

- *Page 14, line 29: please change "Figs. 5-11" for "Figs. 5-9 and 11". Moreover, is Fig. 10 done with the data from 0 to 24h or with data from 9 to 19h?*

Reply : The list of figures has been adapted. All plots contain only data from 9 - 19 UTC. We state this more clearly now in section 2 model and data (p. X5 l. 15/16): "All simulations are run for 24 h. If not stated otherwise, the analysis presented in this paper focusses on the time period between 9 and 19 UTC, i.e. the time period of main convective activity."

- *Page 14, line 30: the variables are not plotted in a box-plot but in an error bar type plot.*

Reply : Yes, of course. Thank you for pointing this out.
 The sentence has been corrected (p. 14, l. 20-24): "If instantaneous realisations of the different (domain-averaged) variables would be considered (box-plots on left side of the figures), the variability would be much larger than suggested by the domain mean plots (right side of the plots)."

- *Page 14 line 35 and page 15, line 1: could it be related to a longer cloud lifetime?*

Reply : We are not sure what the reviewer is referring to. Is it possible that this comment refers to p. 15, l. 35/p. 16, l. 1. If so, a longer lifetime could of course influence the frozen fraction as well, but it is not possible to investigate this effect in our simulations, we do prefer not to comment on any cloud lifetime changes.

*Section 7:*
- *Page 16, lines 17-19: I suggest changing the sentence since some of the cloud properties stated here are poorly modified by the aerosol perturbations (e.g. cloud fraction), not modified considering all perturbations (e.g. condensation gain G is not significative for standard-high comparison), and not modified if unpaired cases are considered (e.g. precipitation rate or cloud fraction are only significative for paired cases)*

Reply : The sentence has been modified to: "Changes in aerosol concentrations can potentially modify cloud field properties, e.g. cell number and size, cloud depth, cloud fraction, and the domain-wide condensate budget (condensate gain and loss, precipitation rate)." (p. 16, l. 14)

- *Page 16, lines 22-34: Could you explain in more detail how the significance analysis of paired and unpaired was done? How do you pair the ensemble members? I do not really see the advantage of the paired significances with so few ensemble members and it looks to me confusing, if not misleading, since as you already say in page 16, lines 26-27, the statistical analysis is based on a very small sample, which affects the validity of several assumptions. Therefore, I personally prefer the results with unpaired cases because the sample is already too small to be paired and because the results with unpaired cases better reflect the results and error bars (spread) shown in all figures and in particular in Fig 12, even at the expense of having less significative results.*

Reply : The ensemble members are paired according to their meteorological initial conditions, i.e. are computed by assuming at the cloud properties for the three simulations with different aerosol but identical meteorological initial conditions are not independent. The statistical analysis of paired ensemble members tests the significance of aerosol-induced changes, if aerosol perturbations are considered for identical meteorological conditions but for a number of cases. Hence this reflects the approach taken by most previous modelling studies. We think the difference in the significance between the unpaired and paired ensemble is interesting and may offer an explanation as to why ACI is often found more pronounced in model, case-study based analysis as compared to observational studies.
We agree that the sample size is an issue in the presented work, which is aggregated by the pairing of ensemble members for statistical analysis. However, we decide to still use show this data, as it is consistent with the physical analysis of changes and therefore we think broad picture painted by the statistical analysis is not severely affected by sample size issues.
We modified the introductory text in section 5 (p. 10, l. 22-27):
"In this section, we compare the aerosol signal in the different meteorological ensemble members, i.e. the difference in realisations with different aerosol scenarios but identical aerosol initial and boundary conditions. Thereby we test the robustness of aerosol-induced changes to small perturbations in the meteorological conditions. To quantify the significance of aerosol-induced changes we use a two-sided t-test for ensemble members paired according to meteorological conditions (Table. 1). Using paired ensemble members reflects the interdependence of cloud properties in realisations with different aerosol but identical meteorological initial and boundary conditions."
And in section 7 (p. 16, l. 20-22):

"First, the idealised situation where the meteorological initial conditions are identical for different aerosol perturbations is assessed by pairing ensemble members according to the meteorological initial conditions. This is equivalent to testing the statistical significance of the differences between realisations with different aerosol scenarios and identical meteorological initial and boundary conditions.

*Page 18, lines 31-32: could you give an example of those "variables closely related to aerosol concentrations" and for those "variables that are linked to aerosol concentrations by a series of complex processes" which apply to the investigated case?*

Reply : Yes certainly. New text: "Consistent with previous studies, we find that the aerosol signals in variables closely related to aerosol concentrations, such as for example cloud droplet number concentrations, are easier to retrieve than for variables that are linked to aerosol concentrations by a series of complex processes, such as for example accumulated surface precipitation." (p. 18, l. 32-34)

*I would suggest changing the section 4, 5 and 6 titles since they are not helpful for understanding the article structure. In my opinion, it would be easier for the reader if they are rewritten somehow in that way:*
- *Section 4: Comparison of the cloud-properties results among 10 different meteorological ensemble members (unperturbed aerosol profiles)*
- *Section 5: Analysis of the results regarding aerosol-induced changes (3 aerosol concentration scenarios) among 10 ensemble members*
- *And section 6: Cloud-adjustments attribution (due to initial meteorological and boundary conditions or aerosol concentration loads)*

Reply : We agree that the previous section titles were not ideal. They have been altered to:
Section 4: Cloud property variability in the meteorological ensemble (standard-aerosol scenario only)
Section 5: Cloud property changes between ensemble members with different aerosol and identical meteorological initial and boundary conditions
Section 6: Contribution of aerosol and meteorology perturbations to overall cloud property variability

*References:*
- *Fan et al. 2016: has the DOI link repeated.*
- *Sheffield et al. 2015: has the DOI reference repeated?*
- *Tao et al. 2012: please check if the reference between the DOI and the year should be there.*

Reply : Thank you for finding these. The references have been corrected.

*Figures:*
*General comments on the figures:*
*Generally speaking, the way the figures are presented is a bit chaotic. First of all, they are not correctly ordered, and secondly there are too many. I suggest the following improvements:*
- *Re-organize all the figures in order of appearance in the text.For example: figures 6 and 7 are referenced in the text before 4 and 5 have been, and figure 11 before figure 10.*

Reply : Figures have been re-order to match their mentioning in the text. Note that some figures have been moved to the SI and the order they are mentioned in the text has also be somewhat changed.

- *Remove linking lines between ensemble members from the following figures: 4b, 7a and b, SI 8a, and SI 9.*

Reply : Done.

- *Consider changing the color palette for the different aerosol load runs in the following figures, since it is difficult to differentiate them: 4b, 7, SI 8a, and SI 9.*

Reply : We refrain from changing the colour palette, as it is identical to the one used in the already published first part of the manuscript. However, to make it more easy to distinguish the differences between aerosol scenario, we have supplied difference plots (low / high scenario - std scenario).

*Comments on figures from the main discussion paper:*
- *Fig. 1:*
  - *In the caption it is stated that 9 ensemble members were chosen from 33. Could you give more information on that? How they were chosen? Which criteria were used?*

    Reply : The selection of ensemble members for downscaling is described in section 2 (p. 4, l. 27-32). We added a reference to this description in the caption.

  - *Could you change the x-axis ticks in a way that both graphs (a and b) have the same.*

    Reply : Done.

- *Fig. 3: please add the data information in the last column "mean" in c and d, otherwise remove it from the graphs.*

Reply : Columns have been removed.

- *Fig. 4: consider removing it or moving it to the SI since it is only cited once in the paper (and actually only Fig. 4a), the latent heat flux (Fig. 4b) is not used in for the discussion, and the sensible heat flux figure is only used once. Moreover, I think the caption is wrong because it does not match with any of the three graphs and legends.*

Reply : Figure 4 b and c have been removed from the manuscript, as they are not discussed in the revised text. The caption has been corrected.

- *Fig. 5: since it does not show big differences between ensemble members I would suggest moving it to the SI.*

Reply : We have moved Fig. 5 b to the SI. Fig. 5a remains in the manuscript, as it is the only variable with a clear separation from between aerosol scenarios. This is itself an important point of the manuscript. Also, we think this plots helps the reader to understand the following plots of this type, which are more messy due to the larger meteorology-induced variability.

- *Fig. 7: as said before, I recommend changing the color palette of the graphic or enlarging the figure.*

Reply : We refrain from changing the colour palette, as it is identical to the one used in the already published first part of the manuscript. However, to make it more easy to distinguish the differences between aerosol scenario, we have supplied difference plots (low / high scenario - std scenario).

- *Fig. 10: please change the legend with the symbols that appear in the graph (i.e. there are no squares in the graph). Also the caption is wrong since there are no "black symbols", "downward pointing triangles" or "upward pointing triangles".*

Reply : Legend has been modified and the text in the caption has been corrected.

- *Fig.12: this is a really interesting and helping figure. just want to say that adding a legend or a caption explanation regarding the colors, as well as regarding the acronyms used in the x-axis, would improve it. Please, also include if CDNC is at the cloud base or at the top.*

Reply : Thank you for the positive comments on this figure. The additional information has been included in the caption.

*Comments on the SI figures:*
- *SI Fig. 1: I would include this figure in the main paper (not in the supplement) since it helps the reader quickly identifying the region on the model simulations have been done and how they look like.*

Reply : This figure is now in the main paper (new Fig. 2).

- *SI Fig. 4: it needs some improvements since it is not intuitive. I would suggest plotting in different colors the temperature and the dew temperature profiles, from both model and observational data.*

Reply : The suggested improvements have been made.

- *SI Fig 6 and 7 captions: "The box plots represent the temporal variabilitiy of each variable" should read "The box plots represent the temporal variability of the variable" or "The box plots represent the temporal variability of the variable for each ensemble member".*

Reply : Done.

- *SI Fig. 6: why ensemble 4 with low aerosol look so different from the others? Is there any apparent reason?*

Reply : Thanks for spotting this. There was an error in the plotting routine affecting in particular this specific ensemble member. The plot has been corrected.

- *SI Fig. 7: CAPE for cntl simulation is missing. And please, either add values for the last column or remove "mean" from the graphic.*

Reply : Thanks for spotting this. The plot has been corrected.

- *SI Fig. 9: I suggest adding into the caption the description of IG and IL (from the legend) as well as "[...] condensation (C) and deposition rate (D) [...]".*

Reply : The caption as well as the plot itself have been improved and the inconsistent/unclear notation have been cleaned up.

*Tables:*
- *Table 1: as mentioned before, I would remove the unpaired results.*

Reply : see reply to comment on Page 16, lines 22-34

*Typos:*
*Page 4, line 17: "section ??" should read "section 5".*
*Page 7, line 24 and line 28: "15 UTC" should read "15.20 UTC".*
*Page 10, line 17: "coherent areas" or "consecutive areas"?*
*Page 10, line 18: "arial fraction" should read "areal fraction".*
*Page 11, line 9: the word "and" is missing between "lower" and "respectively".*
*Page 11, line 13: the word "part" is missing between "first" and "of this study".*
*Page 12, line 5: remove "is" from the sentence.*
*Page 13, line 10: remove "also" from the sentence.*
*Page 13, line 32: add the word "by" after "cloud top and".*
*Page 13, line 34: change "increasing" by "increased".*
*Page 14, line 11: "section" should read "sections".*
*Page 18, line 21: "aerosol-induce" should read "aerosol-induced".*

Reply : Thank you very much for pointing these out! We corrected all issues as suggested.

---

## Referee Report (RR1)

P6 L21-22: "Nevertheless, previous evaluation studies of convection-permitting ensemble simulations have also reported precipitation forecasts to be underdispersive over longer evaluation periods (e.g. Romine et al., 2014; Schwartz et al., 2014) "

P7 L24: "but the radiosonde passed through clouds"

P10 L16: "the variations… is dominating" -> "the variations… dominate"

P 13 L29: you have a missing figure reference (Fig. reffig:cdnc )

SI P5 Fig 4: "red (orange) curves"

SI P8 Fig 7a: On screen I find it very hard to distinguish between the colors used for 12 UTC and for 14 UTC. (Fig 7b looks much better).

SI P9 Fig 9b: what do the sizes of the circles represent?

SI P10 Fig 10: "variability"

SI P11 Fig 11c: what do the sizes of the circles represent?

---

## Author Response (AR2)

**Reply to RC 1**

*P6 L21-22: "**Nevertheless,** previous evaluation studies of convection-permitting ensemble simulations have* *also* *reported precipitation forecasts to be underdispersive over longer evaluation periods (e.g. Romine et al., 2014; Schwartz et al., 2014) "*
*P7 L24: "but the radiosonde passed through clouds"*
*P10 L16: "the variations... is dominating" -> "the variations... dominate"*
*P 13 L29: you have a missing figure reference (**Fig. reffig:cdnc** )*
*SI P5 Fig 4: "red (orange) curves"*

Reply: Corrected as suggested. Thank you!

*SI P8 Fig 7a: On screen I find it very hard to distinguish between the colors used for 12 UTC and for 14 UTC. (Fig 7b looks much better).*

Reply: The new version of the plot uses the same color scale as Fig. 7b.

*SI P9 Fig 9b: what do the sizes of the circles represent?*
*SI P11 Fig 11c: what do the sizes of the circles represent?*

Reply: The sizes of the circles in both Figures only vary to make it easier to distinguish points representing the same percentiles. We have clarified this in the caption.

*SI P10 Fig 10: "variability"*

Reply: Done.

**Reply to RC 2**

*I am concerned because it seems that the authors have not corrected some of the errors found by the two reviewers in the first review iteration process. For instance, there are still some typos in the figure numbers, and there are still figures with lines connecting the dots of different ensemble members (which we agreed has no sense and that must be removed, e.g. Fig. 9a).*

Reply: We have removed the lines in Fig. 9a now. The wrong figure references on p. 10 l. 11 and p. 13 l. 14 have been corrected.

*Besides, this review iteration has been wearisome due to the fact that most of the pages and lines referenced in the answer to reviewer document are not correct (from the 12th answer until the end, the reference to page and line number are wrong), and because some of the changes in the main paper are not marked in green (e.g.*
*p. 2, l. 7: "received particular attention"*
*p. 2, l. 8: "strongly over the historic period"*
*p. 5, l. 6: "0000 UTC"*
*p. 5, l. 8: "horizontal"*
*p. 5, l. 19: "in this study is configured"*
*p. 9, l. 15: "4.2. Cloud-property variability" (title)*
*p. 9, l. 24: "(e.g., Fridlind et al., 2010)*
*p. 9, l. 25: "areal" and "(e.g., Grosvenor et al., 2017)"*
*p. 9, l. 27: "However"*
*p. 10, l. 25: "respectively"*
*p. 36, l. 1: "0900")*
*and none of the changes in the supplement, which makes me think perhaps many other changes were not marked neither.*

Reply: Thank you very much for checking the supplied material so carefully! We are really sorry for the inconsistencies, but unfortunately cannot correct these in hindsight.

*On top, the number of figures has not been reduced, as suggested. Instead some of the figures have been combined as sub-figures into single figures, sometimes without any compelling reason.*

Reply: The total number of Figures as given by the figure numbers has not been reduced. However, if sub-panels are included the number of plots has been reduced from 29 (original manuscript) to 19 (revised version). This excludes Fig. 2, which has been added on request of RC 1. The number of figures is similar in both versions mainly because the previous Fig. 5 has been split into three different figures to add new Fig. 8b (on request of RC1) and still have a meaningful set of sub-panels in each figure. Previous Fig. 10 and Fig. 9 b, d have been combined into a single figure, as they are the only figures pertaining to the condensate budget analysis, and to minimise the number of figures in the manuscript.

*Specific issues:*

*Although the authors used the significance results presented in Table 1 all over the article, as I suggested, I still have some considerations to add:*

*- It would be fair to remark that some statements are only valid for the paired ensemble members tests and add "and only for paired ensembles" to the sentences in:*
*p. 11, l. 29-30 ("The change of cloud top height is only significant for an increase in aerosol concentrations from the low to the standard scenario"),*
*p. 12 l. 8-9 ("Aerosol-induced modifications of CR and G are only significant for a decrease of aerosol concentrations relative to the standard scenario").*
*p. 14, l. 5 ("Aerosol-induced modifications to the outgoing radiative fluxes are significant at the 5 % level").*

Reply: We do not think this makes sense, because the unpaired test is only introduced in section 6 and all the references given are from section 5. Also, it is very clearly stated in the introduction to section 5 that all statistical results in this section are from paired tests.

*- I think they mean 95 % level of confidence instead of 5 %, since 5% level of confidence would be very low.*

Reply: In the text we only use the significance level (and not the confidence level). The significance level is given correctly as 5 %.

*Figure 1 provided in the "Reply to comments form reviewer #2" has the OLR_cc plot twice (bottom left and middle right), and the capture is not correct, since it refers to the mean cloud fraction (bottom right), which is not shown in the figure. Additionally, I recommend the authors not to label the figures this document in the same way as in the paper but with roman numbers or letters (e.g. Fig I, II… or Fig. a, b...) since it becomes very confusing to have figure numbers for the old version paper, for the new version and for the reply figures.*

Reply: We are sorry for the inconsistency between the caption and the actual figure as well as the confusion caused by similar figure references in the reply document and the manuscript. However, we are not able to correct this in hindsight.

*Furthermore, there are figures wrong referenced in the text, in particular "SI Fig. 12" in page 13 line 14 should read "SI Fig. 13". In fact, SI Fig. 13 is not cited at all in the text (I think it should be cited at least in p. 16 l. 3), as well as SI Fig. 14b and SI Fig. 15b which are not cited neither.*

Reply: We corrected the reference (p. 13 l. 15) and added references to SI Fig. 13 on p. 16 l. 3 and 4, as suggested. Also, SI Fig. 14 b is now referenced on p. 12 l. 4 and p. 15 l. 17. SI Fig. 15 is referenced on p. 15 l. 24.

*Some sentence citations of the altered new version paper text are not correctly written in the answers document ("Reply to comments form reviewer #2"):*
*- You said the altered text in p.10 l. 13-15 l. 18-20 is "[…] (SI Fig. 13)" and it says "[…] (SI Fig. 12)" in the paper text.*
*- You said you renamed section 5 as "Aerosol-induced cloud property changes in different meteorological ensemble members (paired meteorology)", but in fact the new title which appears in the paper text is "Cloud property changes between ensemble members in different aerosol and identical meteorological initial and boundary conditions".*
*- You said in p. 12 l. 21-23 l. 27-29 "[…] Fig. 12", but in fact in the text says "[...] Fig. 9B".*
*- Answering to my question regarding the PE change you refer me to Fig. 13 which it does not exist!*
*- Figure 3 of the "Reply to comments from reviewer #2" is not mentioned anywhere, I guess you identified it as "Fig. 4" in your reply.,*

Reply: We are sorry for these inconsistencies!

*Regarding the answer to my question "Page 13, lines 12-14: I do not think "all percentiles up to and including the 75 th percentile show an increase with the aerosol concentration" for all ensemble members, since in ensemble member 4 the standard aerosol is lower than the other two and in member 7 the high aerosol is lower than the low and standard aerosol concentration", first of all, you answered that the text has been modified but the sentence you wrote in the answer document is not the same written in green in the paper. And secondly, I think the sentence is still not correct, since it is not true for ensemble number 7, consider changing it for "almost all ensemble members".*

Reply: Thank you for pointing this out. We have modified the sentence as suggested (p. 13 l. 7/8).

*After, you answered that "the aerosol direct effect is not included in the model simulations", then, I think it should be stated somewhere in the methodology since it has major effects on the radiative balance latter on.*

Reply: We have added this to the "Data and Methods" section.

*Some comments on the figures:*

*Figure 9:*
*I would change the sentence "The dots represents [...]" for "The dots and diamonds represent [...]".*
*I would change the words "open"/"closed" for "filled"/"unfilled".*
*Please add a filled symbol in the legend referring to "without advective term" simulations to make it complete.*

Reply: Done.

*Figure SI 4: reference to the orange curves is missing. Please, consider adding "(orange)" after "red".*

Reply: Done.

*Figure SI 12: it could be improved by plotting all three variables in a single plot instead of three (with different colors or symbols). Then, the three variables will be in the same x-y axis and the magnitude of each of them will be much more obvious than is now, and therefore better showing how the impact of aerosols in clear sky is prevailing over cloudy skies.*

Reply: Done. Thank you for the suggestion.

*Typos:*

*- p. 4, l. 31: "[…] from each cluster the member closest to the mean cluster […]" should read "[…] from each cluster the closest member to the mean cluster […]".*
*- p. 9, l. 29: reference to non-existing figure 10c should read "Fig. 10".*
*- p. 13, l. 29: "Fig. reffig:cdnc"*

Reply: Done.

*Regarding all the other technical issues and questions I am satisfied with the answers provided by the authors.*

*I recommend the article publication in ACP after minor revision.*

**Aerosol-cloud interactions in mixed-phase convective clouds. Part 2: Meteorological ensemble.**

Annette K. Miltenberger[1], Paul R. Field[1,2], Adrian A. Hill[2], Ben J. Shipway[2], and Jonathan M. Wilkinson[2]

[1]Institute of Climate and Atmospheric Science, School of Earth and Environment, University of Leeds, United Kingdom
[2]Met Office, Exeter, United Kingdom

*Correspondence to:* Annette K. Miltenberger (a.miltenberger@leeds.ac.uk)

**Abstract.** The relative contribution of variations in meteorological and aerosol initial and boundary conditions to the variability in modelled cloud properties are investigated with a high-resolution ensemble (30 members). In the investigated case, moderately deep convection develops along sea-breeze convergence zones over the southwestern peninsula of the UK. A detailed analysis of the mechanism of aerosol-cloud interactions in this case has been presented in the first part of this study (Miltenberger et al., 2018).

The meteorological ensemble (10 members) varies by about a factor of 2 in boundary layer moisture convergence, surface precipitation, and cloud fraction, while aerosol number concentrations are varied by a factor of 100 between the three considered aerosol scenarios. If ensemble members are paired according to the meteorological initial and boundary conditions, aerosol-induced changes are consistent across the ensemble. Aerosol-induced changes in CDNC, cloud fraction, cell number and size, outgoing shortwave radiation, instantaneous and mean precipitation rates, and precipitation efficiency are statistically significant at the $5\,\%$ level, but changes in cloud top height or condensate gain are not. In contrast, if ensemble members are not paired according to meteorological conditions, aerosol-induced changes are statistically significant only for CDNC, cell number and size, outgoing shortwave radiation, and precipitation efficiency. The significance of aerosol-induced changes depends on the aerosol scenarios compared, i.e. an increase or decrease relative to the standard scenario.

A simple statistical analysis of the results suggests that a large number of realisations (typically $> 100$) of meteorological conditions within the uncertainty of a single day are required for retrieving robust aerosol signals in most cloud properties. Only for CDNC and shortwave radiation small samples are sufficient.

While the results are strictly only valid for the investigated case, the presented evidence combined with previous studies highlights the necessity for careful consideration of intrinsic predictability, meteorological conditions, and co-variability between aerosol and meteorological conditions in observational or modelling studies on aerosol indirect effects.

*Copyright statement.* The works published in this journal are distributed under the Creative Commons Attribution 3.0 License. This licence does not affect the Crown copyright work, which is reusable under the Open Government Licence (OGL). The Creative Commons Attribution 3.0 License and the OGL are interoperable and do not conflict with, reduce, or limit each other.

[revised manuscript text omitted]

 (20, 60) | 420
 (290, 790) | 60
 (40, 90) |